# The nitrogen starvation-induced inhibitor Rts3 restrains Sit4/PP6 to gate quiescence downstream of TORC1

Ladislav Dokládal [1], Marie-Pierre Péli-Gulli [1], Josephine Alba [1,3], Michael Stumpe [1], Malika Jaquenoud [1], Insa Klemt [2], Cyril Andrea Jaggi [1], Rebecca Lourdes Calviello [1], Zehan Hu [1,4], Devanarayanan Siva Sankar [1,5], Matthias Peter [2], Jörn Dengjel [1] & Claudio De Virgilio [1] ✉

Cellular quiescence is a reversible state essential for survival under nutrient-limiting or growth-restrictive conditions, yet the mechanisms fine-tuning its depth and reversibility remain elusive. Here, we identify *Saccharomyces cerevisiae* Rts3 as a regulator of the quiescence trajectory downstream of TORC1. Using phosphatase inhibitor beads and mass spectrometry, we characterize Rts3 as a phosphatase interactor in rapamycin-treated cells and define it as an inhibitor of the PP6 phosphatase Sit4. Mechanistically, it employs an α-helix to dock directly into the Sit4-Sap185/190 catalytic cleft. Transcriptionally induced by Gln3/Gat1 during nitrogen starvation, Rts3 is rapidly degraded upon nutrient repletion via a TORC1-SCF$^{Cdc4}$-proteasome axis. By selectively constraining Sit4-Sap185/190 activity, this inhibitor modulates nitrogen-responsive transcriptional and translational programs to prevent excessive accumulation of Gln3/Rtg3 targets, establishing a feedback loop gating quiescence depth to support long-term survival. Our findings position Rts3 as a dynamic molecular brake on Sit4, ensuring a protective yet reversible quiescent state.

Eukaryotic organisms have developed complex signaling networks to perceive and react to dynamic environmental changes. This ability is crucial for the precise regulation of key cellular processes such as growth, metabolism, and proliferation. A highly conserved hub within this regulatory framework is the Target of Rapamycin Complex 1 (TORC1), a central regulator of cell growth and metabolism, which integrates various signals, including those from growth factors, hormones, and nutrients, especially amino acids[1,2]. Upon activation, TORC1 stimulates anabolic processes, such as protein synthesis, while simultaneously suppressing catabolic processes like autophagy. The critical role of TORC1 is underscored by its association with numerous human diseases, including cancer, obesity, and neurodegenerative

disorders, when its regulation is impaired, highlighting its profound influence on overall cellular and organismal health[3].

In the budding yeast *Saccharomyces cerevisiae*, TORC1 critically controls the balance between cell proliferation and the transition into a quiescence program that is characterized by reduced metabolic activity, enhanced stress resistance, and specific changes in gene expression, which are vital for the long-term survival of yeast under challenging environmental conditions[4,5]. Key events downstream of TORC1 involve type 2 protein phosphatases (PP2As; i.e., Pph21 and Pph22) and Sit4 (PP6 in mammals)[6,7]. A subset of these enzymes interacts with, and is regulated by, TORC1 through Tap42 and Tip41[6,8–12]. Sit4 activity is suggested to be enhanced by Tip41-mediated

[1]Department of Biology, University of Fribourg, Fribourg, Switzerland. [2]Department of Biology, Institute of Biochemistry, Eidgenössische Technische Hochschule (ETH), Zürich, Switzerland. [3]Present address: Biologics Research Center, Novartis Campus, Basel, Switzerland. [4]Present address: PreOmics GmbH, Planegg/Martinsried, Germany. [5]Present address: Global Health Institute, Swiss Federal Institute of Technology Lausanne (EPFL), Lausanne, Switzerland. ✉e-mail: Claudio.DeVirgilio@unifr.ch

Tap42 sequestering, which allows Sit4 to interact with Sit4-associated proteins (SAP) instead of Tap42[8]. Consistent with this mechanism, TORC1 influences both transcriptional and translational programs via Sit4-SAP complexes[13]. SAPs include two pairs, Sap185/Sap190 and Sap4/Sap155, that have each arisen from genome duplication, share significant homology among each other, and carry out partially redundant functions in controlling Sit4 activity and specificity while competing with one another for Sit4-binding[14–16]. Upon TORC1 inhibition, the Sit4-Sap185/Sap190 complexes are specifically important for the induction of (i) nitrogen catabolite-repressed (NCR) genes that depend on the transcription factors Gln3 and Gat1[17], (ii) expression of *CIT2* that requires the mitochondrial retrograde transcription factor Rtg3 (combined with Rtg1)[18], and (iii) inhibition of protein translation via the Gcn2 protein kinase as part of the general amino acid control (GAAC) pathway[19]. Sit4-SAP complexes were therefore proposed to dephosphorylate multiple target proteins such as Gln3, Rtg3, and Gcn2 (and possibly also the Npr1 protein kinase and the elongator subunit Elp1)[8,14] to execute the cellular response to nitrogen or amino acid starvation[20–23], but the regulatory mechanisms that fine-tune their activities remain poorly understood.

To gain further insight into the mechanisms that shape the protein phosphatase (PPase) landscape downstream of TORC1, we established here an unbiased, comprehensive profiling of PPases and their interacting proteins affected by rapamycin-induced TORC1 inhibition by utilizing an approach combining PPase inhibitor beads and mass spectrometry (PIB-MS)[24]. Rts3 emerged as the sole PPase-associated protein markedly enriched upon rapamycin treatment compared to untreated controls, and our findings uncover Rts3 as a regulator of cellular quiescence. Indeed, Rts3 levels increase under nitrogen starvation, and Rts3 is rapidly degraded upon nitrogen readdition, a dynamic tightly governed by TORC1 activity and SCF$^{Cdc4}$-mediated ubiquitination. Contrary to prior annotations, Rts3 does not associate with the canonical PP2A complex; instead, it functions as a direct inhibitor of Sit4, engaging its catalytic cleft in Sap185- or Sap190-associated Sit4 complexes. Through this inhibition, Rts3 dampens Sit4 activity during starvation, suppressing the unrestrained accumulation of Gln3- and Rtg3-dependent transcripts that could otherwise promote excessive quiescent depth. Rts3 thus operates within a negative feedback circuit in which Sit4-dependent Gln3 activation induces Rts3 expression, establishing a rheostat that fine-tunes quiescence and enables timely reactivation upon nutrient replenishment.

## Results

### Rts3 is a rapamycin-responsive PPase interactor

To achieve an unbiased and comprehensive profiling of PPases and their interacting proteins affected by rapamycin-induced TORC1 inhibition, we utilized a PIB-MS approach. This technique employs microcystin-LR (MC-LR)-conjugated beads to selectively enrich PPase catalytic subunits and their associated proteins, followed by quantitative mass spectrometry (Fig. 1a, Supplementary Data 1)[24]. Applying this methodology like in mammalian cells[25], we successfully captured all MC-LR-sensitive catalytic subunits of PPase 1 (PP1), 1-like (PP1-like), 2 A (PP2A), 2A-like (PP2A-like), 3 (PP3), 4 (PP4), 5 (PP5), and 6 (PP6), along with select associated regulatory subunits, from rapamycin-treated cell extracts (Fig. 1b, c)[26]. To visualize the functional connectivity of the captured interactome, we mapped the enriched proteins onto a STRING interaction network (Fig. 1c), which confirms the recovery of multiple distinct phosphatase holoenzymes (e.g., PP2A, PP6, and PP1 complexes).

Notably, Rts3 emerged as the sole protein phosphatase-associated protein significantly enriched in rapamycin-treated samples compared to empty beads controls (Fig. 1b). Rts3 is a poorly studied protein that may potentially regulate PPases. Accordingly, genetic studies have identified *RTS3* as a dosage suppressor of caffeine sensitivity and suggested that it may negatively regulate Sit4-SAP

modules[27]. Given that TORC1 is the primary target of caffeine in yeast[28], these findings implied a role for Rts3 in the TORC1 pathway, a notion also supported by the rapamycin-sensitive phenotype observed in *rts3Δ* mutants[27,29]. In addition, large-scale interactome studies also pointed to a more direct role for Rts3 in Sit4 regulation, as it has been identified as an interactor of the dimeric Sit4-Sap185/Sap190 complexes[30,31]. Interestingly, these same studies also suggested that Rts3 might function as a B-subunit within a canonical heterotrimeric PP2A complex, similar to how Rts1 and Cdc55 B-subunits associate with the catalytic subunits Pph21 or Pph22 and the scaffold protein Tpd3[32]. Collectively, while both genetic and interactome data connect Rts3 to the regulation of Sit4 and possibly PP2A, its precise in vivo function and its physiological role in nutrient signaling have remained largely elusive.

### Dual transcriptional and proteasomal control of Rts3

To delineate the Rts3 function, we first quantified its levels using a functional N-terminally tagged GFP-Rts3. Rts3 was barely detectable in exponentially growing cells but was robustly induced upon rapamycin treatment or nitrogen starvation (Fig. 2a). *RTS3* transcript levels changed in a manner that mirrored protein dynamics (Fig. 2b), indicating that the acute induction and shutdown of Rts3 are driven, in part, by transcriptional control. Large-scale promoter annotations and motif analyses suggested that the *RTS3* promoter contains sites for multiple stress- and nutrient-responsive transcription factors[33]. Our targeted deletion analysis, however, pinpointed the GATA factors Gat1 and Gln3 as the primary drivers of *RTS3* induction: loss of both factors substantially blunted rapamycin-triggered *RTS3* upregulation, whereas loss of Gcn4, or Rtg3, or the combined loss of Msn2, Msn4, and Gis1 had no discernible effect (Fig. 2c). These findings align with the established role of Gln3/Gat1 in nitrogen catabolite-responsive transcription downstream of TORC1[34,35]. However, given the residual *RTS3* expression persisting in the *gln3Δ gat1Δ* double mutant, contributions from other, yet unidentified regulatory pathways must exist.

Interestingly, Rts3 abundance was also controlled post-translationally by the ubiquitin-proteasome system, as it was rapidly degraded, within minutes, upon amino acid refeeding (Fig. 2a). Temperature-sensitive mutants in the proteasome (*pup1-1*, a catalytic core subunit mutant; *rpn5-1*, a 19S lid subunit mutant) or treatment with the proteasome inhibitor MG-132 markedly impaired the rapid loss of Rts3 upon amino acid refeeding (Fig. 2d, e). Consistent with the robust accumulation of polyubiquitinated proteins in *rpn5-1* and MG-132-treated cells compared to the delayed response in *pup1-1* (Fig. 2d), the *pup1-1* allele merely slowed Rts3 turnover, whereas conditions imposing a stronger blockade (*rpn5-1* or MG-132) led to the specific accumulation of slower-migrating Rts3 species in SDS-PAGE gels, suggestive of a phosphorylated form that is normally targeted for proteasomal degradation. Co-treatment with MG-132 and rapamycin caused the upper, slower-migrating species of the GFP-Rts3 doublet to disappear, indicating that this species corresponds to a TORC1-dependent phosphorylated isoform of Rts3 (Fig. 2e). TORC1 activity proved decisive for this switch, because rapamycin treatment during amino acid refeeding completely prevented Rts3 degradation (Fig. 2f); in this case, Rts3 accumulated predominantly as a faster-migrating, non-phosphorylated species. To exclude off-target effects, we monitored Rts3 turnover in cells carrying the rapamycin-resistant *TOR1-1* allele. Unlike in wild-type cells (Fig. 2f), rapamycin expectedly failed to stabilize Rts3 in the *TOR1-1* background, where the protein was rapidly degraded upon refeeding regardless of drug treatment (Fig. 2g). Direct phosphorylation of Rts3 by TORC1 was further supported by in vitro kinase assays incubating recombinant His$_6$-Rts3 and TORC1 (immunopurified from yeast) in the presence of radioactive [γ-$^{32}$P]-ATP (Fig. 2h). This phosphorylation was inhibited by wortmannin and stimulated more strongly by Mn$^{2+}$ than Mg$^{2+}$, consistent with previous

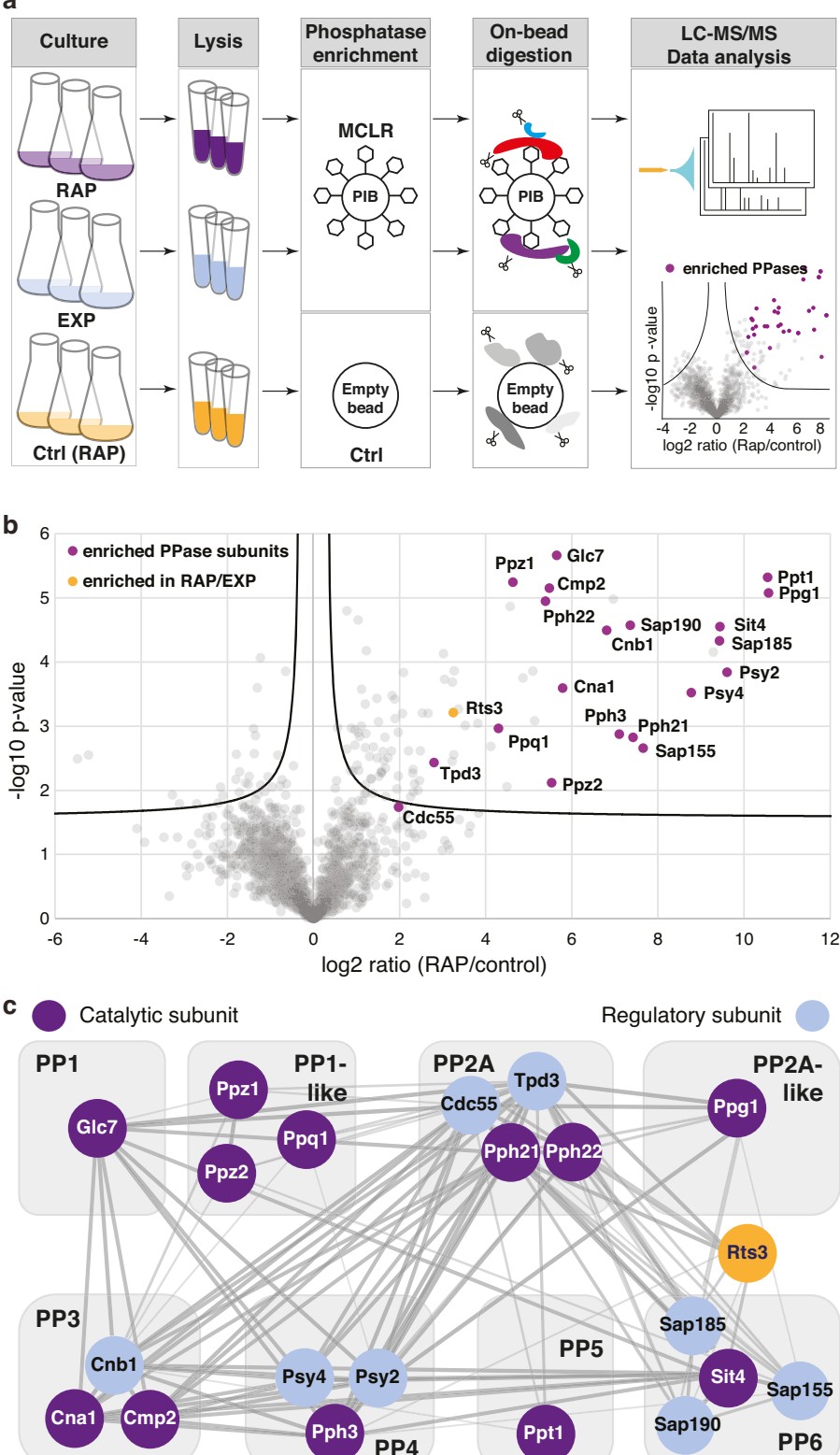

**Fig. 1 | Rts3 associates with protein phosphatases in response to rapamycin.**
**a** Workflow: wild-type yeast cultures ($n = 3$ biological replicates per condition) were lysed and phosphoprotein phosphatases (PPases) enriched using microcystin-LR-coupled Sepharose beads (PIB, phosphatase inhibitor beads). Empty beads served as controls. On-bead digested proteins were analyzed via label-free LC-MS/MS. RAP: rapamycin treatment (200 ng/mL, 30 min); EXP exponential growth conditions. **b** Volcano plot of significantly enriched proteins from rapamycin-treated samples

vs controls ($n = 3$ biological replicates; unpaired two-tailed $t$ test with permutation-based FDR < 0.05). Gray: all proteins; violet: known PPase subunits; orange: Rts3, which shows a robust enrichment (log2 fold-change = 6.5) in RAP vs EXP. The black line indicates the significance cutoff. **c** Interaction network of PPase subunits from (**b**) via STRING DB. Nodes = proteins; edges = interactions (thickness reflects confidence).

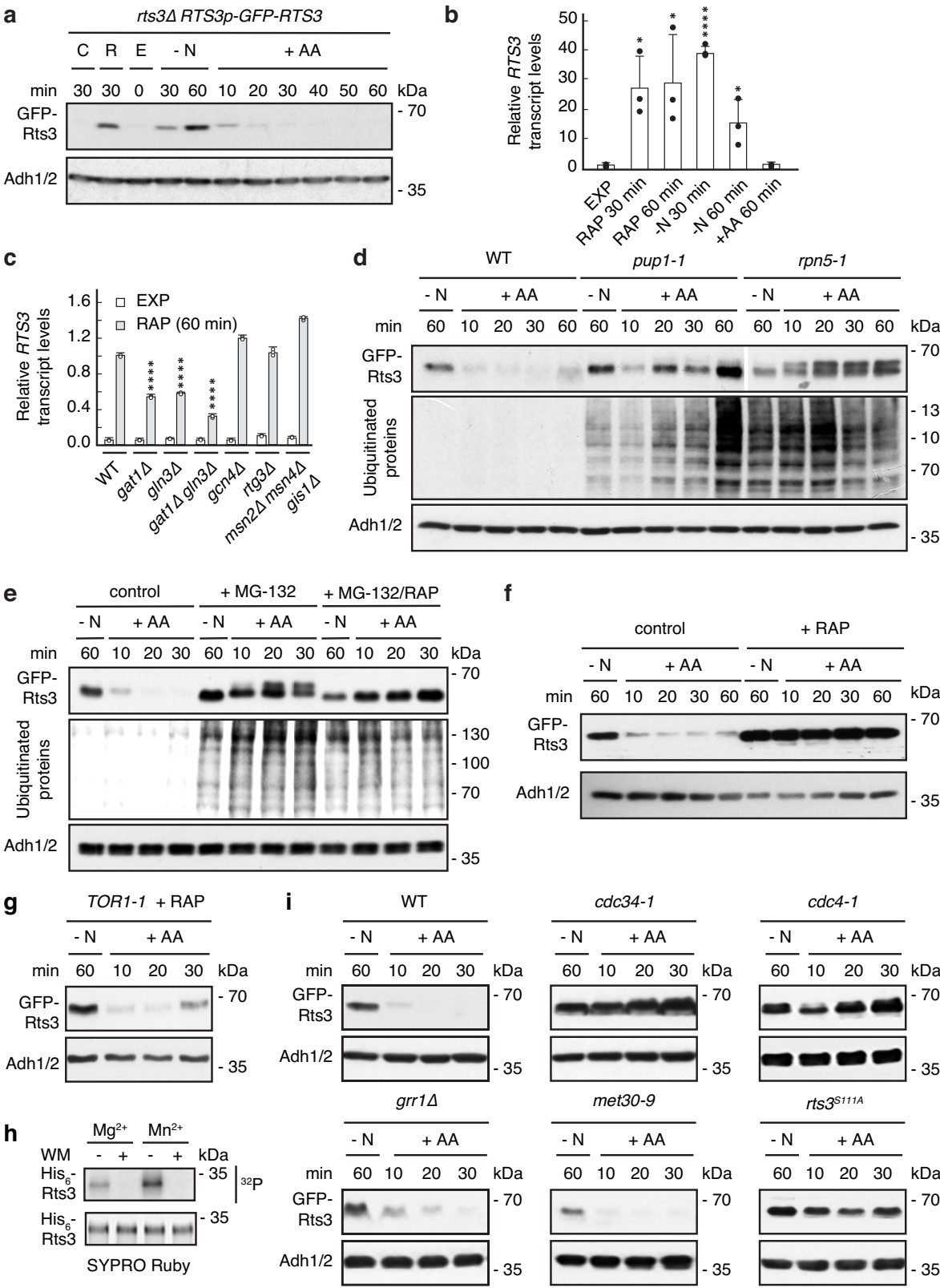

findings that Mn$^{2+}$ is a more effective cofactor for TORC1 activity[36]. We further substantiated this direct connection by detecting a physical interaction between the TORC1 subunit Tco89 and Rts3 using yeast two-hybrid assays, which required stabilization of Rts3 by the proteasome inhibitor MG-132 (Supplementary Fig. 1). Together, these data indicate that TORC1-dependent phosphorylation primes Rts3 for proteasomal degradation upon nutrient re-addition.

Phosphodegron-dependent recognition frequently couples phosphorylation to SCF-mediated ubiquitination[37], and inactivation of the temperature-sensitive E2 enzyme Cdc34-1 indeed stabilized Rts3 during amino acid refeeding (Fig. 2i). Moreover, Rts3 degradation required the F-box protein Cdc4, but not Grr1 or Met30, placing Rts3 under SCF$^{Cdc4}$ control. Consistent with canonical Cdc4 recognition, Rts3 harbors several candidate Cdc4 phosphodegrons that match the

**Fig. 2 | TORC1 controls Rts3 protein levels. a** Rts3 increases upon rapamycin treatment or nitrogen starvation. GFP-Rts3-expressing cells were grown exponentially (E) in synthetic dextrose complete medium without ammonium sulfate (SDC-AS), treated with vehicle (C), rapamycin (R; 200 ng/mL), or nitrogen-starved (-N) and refed with amino acids (+ AA) for the indicated times. Lysates were immunoblotted (anti-GFP; anti-Adh1/2). **b, c** *RTS3* is induced upon rapamycin treatment and nitrogen starvation in a *GLN3/GAT1*-dependent manner. Relative *RTS3* transcript levels in wild-type (WT) (**b**) or mutant (**c**) strains treated as in (**a**) were normalized to *TBP1* (*n* = 3 biological replicates, data shown as mean +SD). Statistics: unpaired two-tailed *t* test vs EXP (**b**) or corresponding WT sample (**c**); for **b** EXP vs RAP 30 min ($p = 0.011$), RAP 60 min ($p = 0.0404$), -N 30 min ($p \leq 0.0001$), -N 60 min ($p = 0.0339$); for **c**, all RAP samples: WT vs *gat1Δ*, *gln3Δ*, and *gat1Δgln3Δ* ($p \leq 0.0001$); * $p < 0.05$, **** $p \leq 0.0001$. **d** Rts3 is stabilized in temperature-sensitive proteasome mutants *pup1-1* and *rpn5-1* upon amino acid refeeding (+ AA) of nitrogen-starved cells (-N). Cells were shifted from 24 °C to 37 °C (1.5 h) prior to nitrogen starvation (-N; 1 h); lysates were immunoblotted using anti-GFP, anti-ubiquitin, and anti-Adh1/2. A different exposure time was used for *rpn5-1* extracts to

match intensities. **e** MG-132 protects Rts3 from degradation upon refeeding (+ AA). During starvation, 75 μM MG-132, 0.003% SDS, and 0.5% DMSO were added; rapamycin (+ RAP; 200 ng/mL, 30 min) was applied where indicated. Analyzed as in (**d**). **f** Rapamycin prevents Rts3 degradation. Cells were starved (-N; 1 h) and refed (+ AA); where indicated, rapamycin (+ RAP; 200 ng/mL) was added after 30 min of N-starvation. Analyzed as in (**a**). **g** *TOR1-1* suppresses rapamycin-induced Rts3 stabilization. Rts3 turnover was analyzed in *TOR1-1* cells upon amino acid refeeding (+ AA) in the presence of rapamycin (+ RAP; 200 ng/mL), as in (**f**). **h** TORC1 phosphorylates Rts3 in vitro. Recombinant His₆-Rts3 was incubated with purified TORC1 ± wortmannin (WM). SYPRO Ruby and autoradiography (³²P) blots are shown. **i** Rts3 is stabilized in temperature-sensitive SCF^Cdc4 mutants (*cdc34-1* and *cdc4-1*) and the *rts3^SIIIA* phosphodegron mutant upon amino acid refeeding (+ AA) of nitrogen-starved cells (-N). Cells were pre-grown at 30 °C (*grr1Δ* and *rts3^SIIIA*) or 24 °C (others), then shifted to 37 °C for 1 h (*grr1Δ* and *rts3^SIIIA* maintained at 30 °C). Analyzed as in (**a**). The experiments (**a; d–i**) were performed three times independently with similar results; representative immunoblots are shown.

respective consensus S/T-P-x-x-(S/T/E/D) sequence[38,39]. Mutational analysis of these candidate sites pinpointed Ser^III as the critical determinant for turnover; the Rts3^SIIIA mutant was fully stabilized upon amino acid refeeding (Fig. 2i). Notably, this stable mutant migrated exclusively as a faster-migrating species, lacking the phosphorylated isoform that typically accumulates when proteasomal degradation is blocked (see Fig. 2d, e). This supports a model in which TORC1-dependent phosphorylation of Rts3-Ser^III triggers SCF^Cdc4-mediated ubiquitination and subsequent proteasomal degradation of Rts3. In sum, TORC1 regulates Rts3 expression both transcriptionally by Gln3/Gat1 in response to TORC1 inhibition and nitrogen stress, and post-translationally by a TORC1-SCF^Cdc4-proteasome-dependent mechanism that accelerates Rts3 clearance upon nutrient restoration. This switch-like mechanism prevents Rts3 engagement during exponential growth and starvation recovery, but allows Rts3 function during starvation, thus ensuring tight temporal control aligned with cellular nutrient status.

## Rts3 modulates TORC1 signaling independently of PP2A

Given prior annotations suggesting that Rts3 might act as a B-type regulatory subunit within heterotrimeric PP2A complexes (with catalytic subunits Pph21/Pph22 and scaffold Tpd3; see above), we first tested these interactions in vivo by co-immunoprecipitation under conditions where Rts3 is highly expressed (rapamycin, 30 min). Surprisingly, while the canonical PP2A B subunits Rts1 and Cdc55 co-immunoprecipitated Pph21/Pph22 and Tpd3 (Fig. 3a), we detected no interaction between Rts3 and these PP2A catalytic core subunits (Fig. 3a). Importantly, the anti-GFP immunoblot confirms that Rts3 was efficiently immunoprecipitated (Fig. 3a), ruling out low bait abundance as a cause for the lack of binding. Furthermore, as described in the subsequent section, this same construct remains fully functional for interaction with Sit4-SAP complexes. These data argue against a stable association of Rts3 with canonical PP2A.

We next revisited the genetic relationship between Rts3 and TORC1. Consistent with prior reports, *rts3Δ* cells were hypersensitive to rapamycin (Fig. 3b)[27,29]. Conversely, overexpression of *RTS3* from the strong *ADH1* promoter, but not expression from its endogenous promoter, conferred rapamycin resistance, indicating that Rts3 dosage antagonizes the effects of TORC1 inhibition (Fig. 3b). Despite these phenotypes, Rts3 did not measurably alter TORC1 kinase output: phosphorylation of Sch9 at Thr^737 and the electrophoretic mobility of Lst4, two established TORC1 activity readouts in yeast[40,41], were indistinguishable between wild-type and *rts3Δ* cells during exponential growth, after rapamycin treatment, under nitrogen starvation, and upon amino acid refeeding (Fig. 3c). These findings place Rts3 functionally within the TORC1 pathway, acting downstream or in parallel to modulate specific effector branches rather than bulk TORC1 activity.

Finally, we examined the subcellular distribution of Rts3 using GFP-tagged Rts3 expressed from its endogenous promoter. As expected, GFP-Rts3 was barely detectable in exponentially growing cells, accumulated in the cytoplasm and nuclei upon rapamycin treatment or nitrogen starvation, and disappeared rapidly upon amino acid refeeding (Fig. 3d). While this dynamic localization pattern does not pinpoint specific molecular functions, it reinforces the notion that Rts3 does not control TORC1 activation per se, but regulates nutrient-responsive cellular processes upon TORC1 inhibition.

## Rts3 binds and inhibits Sit4-Sap185/190 complexes

To define the Rts3 interaction landscape, we performed proximity-dependent biotinylation (TurboID) and quantified enriched prey relative to controls. This identified 381 high-confidence proteins in the Rts3 neighborhood (Fig. 4a, Supplementary Data 2). A global Gene Ontology (GO) enrichment analysis of these candidates revealed a broad functional engagement with ribosome biogenesis, translation machinery, and ribonucleoprotein complex assembly (Fig. 4b). Within this broad landscape, we specifically noted a distinct enrichment of PPase-associated factors, including Sap185, Sap190, Cmp2, and Cdc55 (Fig. 4a; Supplementary Data 2). Of note, because our co-immunoprecipitation analyses did not support a stable interaction of Rts3 with PP2A, the detection of the PP2A B subunit Cdc55 in the Rts3 proximome likely reflects spatial proximity, suggesting that Rts3 resides in overlapping subcellular territories rather than forming direct assemblies with other PP2A species. In parallel to the PPase-associated factors, we also observed a substantial cohort of translation-associated proteins, including 19 ribosomal proteins within the Rts3 neighborhood (Fig. 4a, b, Supplementary Data 2). The latter was confirmed by experiments demonstrating that Rts3 specifically co-purified with cytoplasmic ribosomes when extracted from nitrogen-starved cells (Supplementary Fig. 2a, b), suggesting an unanticipated link to protein translation and/or ribosome hibernation.

Since our STRING-based network analysis of the PPase-specific subset highlighted the Sit4 subunits Sap185 and Sap190 (Fig. 4a), we then performed co-immunoprecipitation assays in rapamycin-treated and nitrogen-starved cells, where Rts3 expression is elevated. These experiments confirmed a robust association of Rts3 with Sit4, Sap185, and Sap190 under TORC1-inhibited conditions, which is in line with previous reports (Fig. 4c)[30,31]. These data place Rts3 in close proximity to Sit4 regulatory subunits in vivo and prompted a structural model illustrating how Rts3 may engage the Sit4-SAP complexes. Given that Rts3 is largely disordered but contains an α-helix spanning residues 64-103, we employed AlphaFold-based predictions to model the Sit4-Sap185 dimer bound to the Rts3 helix (Fig. 4d). To enhance conformational sampling, we then conducted molecular dynamics (MD) simulations of the complex. Our results reveal that the helix docks into

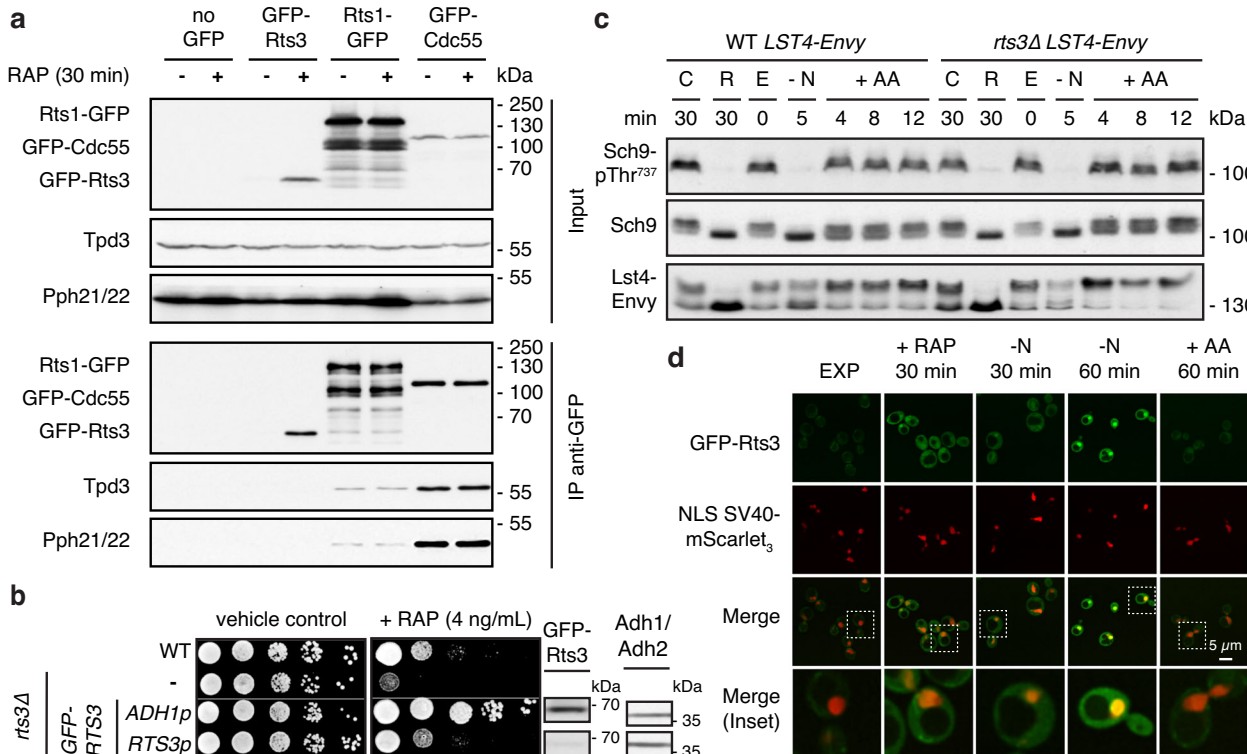

**Fig. 3 | Rts3 modulates TORC1 signaling independently of PP2A association.**
**a** PP2A subunits (Pph21/22, Tpd3) do not co-immunoprecipitate with GFP-Rts3 from native lysates using GFP-trap magnetic agarose. Exponentially growing cells expressing GFP-Rts3, Rts1-GFP, or GFP-Cdc55 were treated or not with rapamycin (RAP; 200 ng/mL). PP2A association with Rts1-GFP and GFP-Cdc55 was used as a positive control. **b** Rts3 modulates rapamycin sensitivity. Strains grown exponentially were spotted (10-fold dilutions) on YPD ± rapamycin (RAP; 4 ng/mL) and incubated for 3 days at 30 °C. Rts3 was expressed from its endogenous promoter or overexpressed from the *ADH1* promoter. GFP-Rts3 expression levels were verified with immunoblotting as in Fig. 2a. **c** Rts3 does not impact phosphorylation of

TORC1 substrates Sch9 and Lst4. WT and *rts3Δ* cells expressing Lst4-Envy were treated as in Fig. 2a for the indicated timepoints and analyzed for Sch9-Thr737 and Lst4 phosphorylation via immunoblotting using anti-pT737-Sch9, anti-Sch9, and anti-GFP. **d** GFP-Rts3 localizes to the cytoplasm and accumulates in the nucleus upon nitrogen starvation. Localization was visualized in *rts3Δ* cells co-expressing SV40-NLS-mScarlet₃ nuclear marker and GFP-Rts3 under its endogenous promoter. Conditions: EXP, RAP (200 ng/mL), nitrogen starvation (-N; 30 and 60 min), and amino acid-refeeding (+ AA) after 60-min nitrogen starvation. Scale bar (5 μm) applies to all images. The experiments (**a**–**d**) were performed three times independently with similar results; representative data are shown.

a groove at the Sit4-Sap185 interface, stabilized by an extensive hydrogen bond network contributed by both partners. Specifically, residues Arg[65], Glu[73], and Asp[77] of Rts3 play key roles in anchoring the helix within the Sit4 groove (Fig. 4e). Crucially, this docking positions the Rts3 helix directly over the catalytic center (e.g., via dynamic hydrogen bonds formed by Asp[77] and, to a lesser extent, Glu[73] with the catalytic residue His[238]; Supplementary Fig. 3a, b), acting as a steric wedge that occludes the active site. This arrangement predicts a mechanism of simple competitive inhibition, where Rts3 physically blocks substrate access rather than altering the intrinsic specificity of the phosphatase. In agreement with this structural prediction, a GFP-Rts3 variant carrying the substitutions R65A, E73A, and D77A (GFP-Rts3[3A]) displayed markedly reduced co-immunoprecipitation with HA-Sit4 and Sap185-V5 compared with wild-type GFP-Rts3 (Fig. 4c). This confirms that these Rts3 residues are essential for anchoring the inhibitor to the Sit4 catalytic cleft. In contrast, its interaction with Sap190-myc₁₃ was less obviously reduced (Fig. 4c). Since the Rts3[3A] mutation disrupts the Sit4 interface (common to both complexes), the residual binding to Sap190 suggests that Rts3 maintains secondary, Sap190-specific contacts that stabilize the complex even when the primary active-site anchor is compromised. Notably, hydrogen bond analysis also identified an interaction between residue Glu[37] of Sit4 and Asn[655] of Sap185 (Fig. 4f), which matches well with the previous observation that mutation of Glu[37] to alanine disrupts the Sit4-Sap185 interaction[14]. In addition, electrostatic surface analysis revealed a positively charged channel along the Sit4-Sap185 cleft, aligning with an

acidic patch on the Rts3 helix (Fig. 4g). This charge complementarity is likely enhancing both the specificity and affinity of the interaction.

Guided by this model, which implies steric occlusion of the active site, we then tested whether Rts3 functions as a Sit4 inhibitor. To this end, we incubated HA-Sit4 (immunopurified from yeast) with increasing concentrations of recombinant His₆-Rts3 (purified from *E. coli*) and assessed phosphatase activity against the synthetic phosphopeptide substrate KRpTIRR. These assays showed that Rts3 progressively suppressed Sit4 activity with an IC₅₀ of 76 nM (Fig. 4h, i). For context, this potency places Rts3 among relatively strong endogenous protein inhibitors of serine/threonine PPases: it is weaker than the most potent inhibitors, such as CPI-17 (IC₅₀ in the low-nanomolar range when phosphorylated at Thr[38])[42], but stronger than regulators like RCAN1, which typically inhibit calcineurin in the high-nanomolar to micromolar range[43]. Consistent with the structural interface, an Rts3[3A] variant targeting residues predicted to form hydrogen bonds within the Sit4 catalytic cleft completely lost inhibitory activity in vitro (Fig. 4h, i). Finally, we assessed functional consequences in vivo. Unlike overexpression of wild-type *RTS3*, overexpression of *rts3[3A]* or *rts3[13A]* (encoding a variant that has amino acids 65–77 replaced by alanines) did not confer rapamycin resistance, confirming that the inhibitory interface is required for function (Fig. 4j). To pinpoint the downstream effector, we overexpressed wild-type *RTS3* in *sit4Δ* cells. We found that resistance was completely abolished in the absence of the phosphatase, establishing *SIT4* as the essential epistatic target. However, unlike the

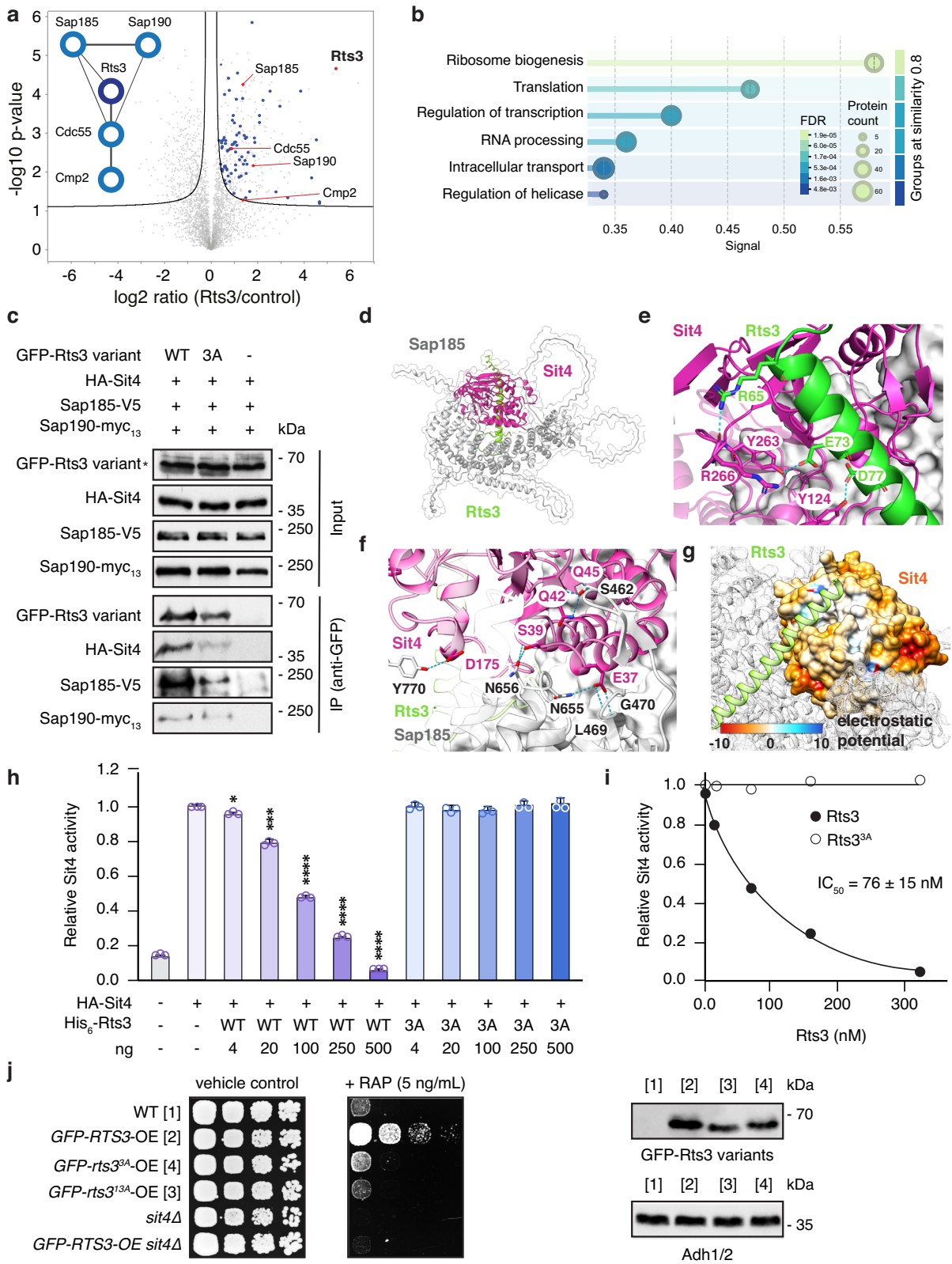

RTS3-overexpression strain, the sit4Δ mutant itself was not rapamycin resistant (Fig. 4j). This distinction aligns with established genetic data showing that rapamycin resistance arises specifically from the loss of the Sap185/190 regulatory branch, whereas the combined loss of Sap4/155 or the catalytic subunit Sit4 does not confer resistance[14]. Thus, RTS3 overexpression does not mimic a total loss of Sit4 activity, but rather phenocopies the specific loss of the Sap185/190

regulatory branch. By specifically restraining the Sit4-Sap185/190 complex, Rts3 likely dampens the toxicity associated with this specific branch while preserving other essential Sit4 functions required for survival. Together, these results identify Rts3 as an endogenous, helix-based inhibitor that docks onto Sit4-Sap185/190 to attenuate Sit4 activity, with structural determinants that are necessary for both biochemical inhibition and TORC1-related phenotypes.

**Fig. 4 | Rts3 binds Sit4-Sap185/Sap190 and inhibits Sit4. a** Rts3 proximal proteins identified via TurboID-V5-Rts3. Volcano plot shows enrichment vs TurboID-V5-Envy control ($n = 3$ biological replicates; unpaired two-tailed $t$ test, FDR < 0.05, $S_O = 0.1$). Black line: significance threshold; gray: all proteins; red: PPase-related; blue: translation-associated. Left: PPase-specific STRING network. **b** GO enrichment of Rts3 proximal proteins (Supplementary Data 2), ranked by STRING signal. Dot size: protein count; color: significance. **c** Rts3 binds Sit4, Sap185, and Sap190 in nitrogen-starved, rapamycin-treated cells. Lysates containing HA-Sit4, Sap185-V5, Sap190-myc$_{13}$, and GFP-Rts3 variants (WT, 3 A [R65A, E73A, D77A]) were analyzed by anti-GFP IP and immunoblotting. Inputs were prepared using native cell lysis (asterisk: non-specific band). Detection utilized light chain-specific secondary antibodies. The experiment was performed three times independently with similar results; representative immunoblots are shown. **d** AlphaFold model of Rts3-Sit4-Sap185. Magenta: Sit4; green: Rts3; gray: Sap185. **e** Hydrogen-bond interactions between Sit4 and Rts3 observed during a 300 ns MD simulation. Key interactions involve Rts3 residues Arg$^{65}$, Glu$^{73}$, and Asp$^{77}$, and Sit4 residues Arg$^{266}$, Tyr$^{263}$, and Tyr$^{124}$. Additionally, Asp$^{77}$ and Glu$^{73}$ form transient hydrogen bonds with active site His$^{238}$ (Supplementary Fig. 3). Magenta: Sit4; green: Rts3. **f** The Sit4-Sap185 interface is stabilized by a hydrogen-bonding network. Sit4 residues precede Sap185 partners: Glu$^{37}$-Asn$^{655}$/Leu$^{469}$/Gly$^{470}$, Ser$^{39}$-Asn$^{656}$, Gln$^{42}$/Gln$^{45}$-Ser$^{462}$, and Asp$^{175}$-Tyr$^{770}$. Magenta: Sit4; gray: Sap185 (black numbering). **g** Electrostatic potential of the Sit4 binding pocket. Negatively charged patches (yellow to orange) stabilize Rts3. **h, i** Rts3 inhibits Sit4 in vitro. Recombinant His$_6$-Rts3 (WT; 3 A) tested in peptide dephosphorylation assays using immunopurified HA-Sit4 from yeast ($n = 3$ distinct samples, data shown as mean +SD; unpaired two-tailed $t$ test vs HA-Sit4 without inhibitor; His$_6$-Rts3: 4 ng ($p = 0.0412$), 20 ng ($p = 0.0002$), ≥100 ng ($p ≤ 0.0001$); * $p < 0.05$, *** $p ≤ 0.001$, **** $p ≤ 0.0001$); IC$_{50}$ was determined via nonlinear regression (GraphPad Prism 10). **j** Rts3 variant overexpression alters rapamycin resistance. Wild-type or *sit4Δ* cells overexpressing WT, 3 A, or 13 A (R65A, E66A, E67A, I68A, I69A, N70A, E71A, M72A, E73A, K74A, E75A, Q76A, D77A) variants of GFP-Rts3 (from the *GPD* promoter) were spotted (10-fold dilutions) and grown for 3 days at 30 °C on SD-Ura plates ± rapamycin (RAP; 5 ng/mL). Expression of Rts3 variants in asynchronously growing cells analyzed by immunoblotting as in Fig. 2a. The experiment was performed three times independently with similar results; representative data are shown.

## Rts3 constrains Sit4 to gate quiescence programs

To test the hypothesis that Rts3 selectively inhibits Sit4 during TORC1 suppression, we performed label-free phosphoproteomics on rapamycin-treated wild-type, *sit4Δ*, and *RTS3*-overexpressing (*RTS3*-OE) cells (Supplementary Data 3). *SIT4* deletion altered 2720 phosphosites across 978 proteins, whereas *RTS3*-OE affected 527 sites in 320 proteins, with 417 phosphosites overlapping in 263 proteins (Fig. 5a). We also profiled *rts3Δ* cells in parallel; however, this dataset yielded few significantly altered phosphosites under stringent statistical thresholds (FDR < 0.05). This likely reflects a detection asymmetry: while *RTS3*-OE imposes a blockade on Sit4 that results in a large, detectable accumulation of phosphorylation (a high signal-to-noise "gain of signal"), the *rts3Δ* mutant enhances the activity of an already-active phosphatase. This creates a "floor effect" where the further removal of residual phosphate groups is difficult to distinguish from the lower limit of detection. Analysis of the robust overexpression dataset revealed that 79% of Rts3-regulated sites are Sit4-dependent, while only 15% of Sit4-dependent sites responded to *RTS3* over-expression, consistent with selective inhibition of the Sit4-Sap185/190 branch rather than global phosphatase suppression. The remaining Rts3-specific sites likely reflect indirect feedback or statistical thresholding rather than off-target activity, as Rts3 variants unable to bind Sit4 failed to elicit any detectable phenotypes (Figs. 4j and 5f).

We next asked which processes are co-regulated by Sit4 and Rts3. GO analysis of shared *sit4Δ*/*RTS3*-OE targets highlighted several categories, including translation and nitrogen utilization (Fig. 5b), prompting us to dissect these signals by mapping them onto protein-protein interaction networks. Focusing first on translation, STRING analysis identified a subcluster of 20 Sit4- and Rts3-dependent phosphoproteins that govern core aspects of protein synthesis (Fig. 5c). Five of these proteins − Pab1, Ssd1, Stm1, Tif3, and Tif4631 − were also present in the Rts3 neighborhood (Fig. 4a, Supplementary Data 2), reinforcing our earlier assumption that Rts3 regulates aspects of protein translation during nutrient stress. The hibernation factor Stm1 controls ribosome homeostasis and is tightly regulated by TORC1-dependent phosphorylation[44,45]. Importantly, *RTS3* overexpression prevented dephosphorylation of Ser$^{41}$ and Ser$^{45}$ in Stm1, while the *rts3Δ* mutant displayed a trend toward reduced phosphorylation at these sites (consistent with Sit4 hyperactivity; Supplementary Data 3). These residues control its association with 80S ribosomes, and thus the balance between active translation and ribosome dormancy. As described above, consistent with the TurboID data, a pulse-chase biotinylation assay confirmed that Rts3 binds mature ribosomes during starvation (Supplementary Fig. 2a, b). However, in contrast to *stm1Δ*, 80S ribosomes assessed by polysome analysis remained stable in starved *RTS3*-OE and *rts3Δ* cells, albeit the levels of disomes were slightly, nevertheless significantly, reduced in *RTS3*-OE cells (Supplementary Fig. 2c, d). To further delineate the physiological range of Rts3 function, we assessed both growth on poor nitrogen sources and survival during prolonged starvation. Given that Sit4 activates Gln3-dependent nitrogen permease expression, we tested growth on proline, which requires the Gln3-regulated transporter Put4[46]. Consistent with Rts3 limiting Sit4 activity, *RTS3*-overexpressing cells exhibited a mild growth defect on proline, whereas *rts3Δ* cells displayed a modest growth advantage (Supplementary Fig. 2e). In addition, *rts3Δ* cells exhibited a significant reduction in chronological lifespan (Supplementary Fig. 2f). These findings indicate that Rts3 fine-tunes nutrient uptake during growth and, consistent with its regulation of Stm1 and ribosome hibernation, plays a crucial role in maintaining viability during starvation.

In parallel with protein translation, we examined a nitrogen utilization network encompassing the transcription factors Gln3, Rtg3, and Ume6, as well as Tip41, the kinase Npr1, and the 14-3-3 protein Bmh1 as shared Sit4- and Rts3-regulated network components (Fig. 5d). To link these signaling changes to output, we next conducted whole-proteome profiling following 3 h of nitrogen starvation. When filtering for proteins induced in a Sit4-dependent manner and opposed by *RTS3* overexpression, targets of Gln3 and Rtg3 were enriched (Fig. 5e, Supplementary Data 4). Accordingly, loss of Rts3 enhanced expression of multiple Gln3-controlled genes and the prototypic Rtg3 target *CIT2*, whereas *RTS3*-OE and *sit4Δ* antagonized their accumulation. In addition, several proteins linked to Gcn4 and Fkh1/2 programs (*Saccharomyces* Genome Database [SGD] annotations) were differentially expressed, indicating that the Rts3-Sit4 axis broadly attenuates starvation-responsive transcriptional networks that coordinate metabolic remodeling. To functionally interrogate Rts3 involvement in the GAAC pathway impinging on Gcn4[19,47], we assessed growth on 3-amino-1,2,4-triazole (3-AT), which inhibits histidine biosynthesis and activates Gcn2-eIF2α-Gcn4 signaling. Accordingly, growth on 3-AT serves as a sensitive cellular readout of Gcn4-dependent transcription. In agreement with a role for Rts3 in dampening GAAC signaling, *RTS3*-OE strain exhibited mild 3-AT sensitivity (Fig. 5f), consistent with reduced Gcn4 activity under amino acid stress. Importantly, this phenotype strictly requires the Rts3-Sit4 interface, as overexpression of the Sit4-binding defective *rts3$^{3A}$* or *rts3$^{13A}$* variants failed to sensitize cells to 3-AT (Fig. 5f).

Finally, to clarify the functional positioning of Rts3 within the TORC1 signaling cascade, we performed genetic epistasis analysis. We reasoned that if the rapamycin hypersensitivity of *rts3Δ* mutants stems from unrestrained (hyperactive) Sit4 signaling, then eliminating Sit4 or its downstream effectors should relieve this toxicity. Consistent with this hypothesis, loss of Sit4, Gln3, Npr1, or Tip41 suppressed the

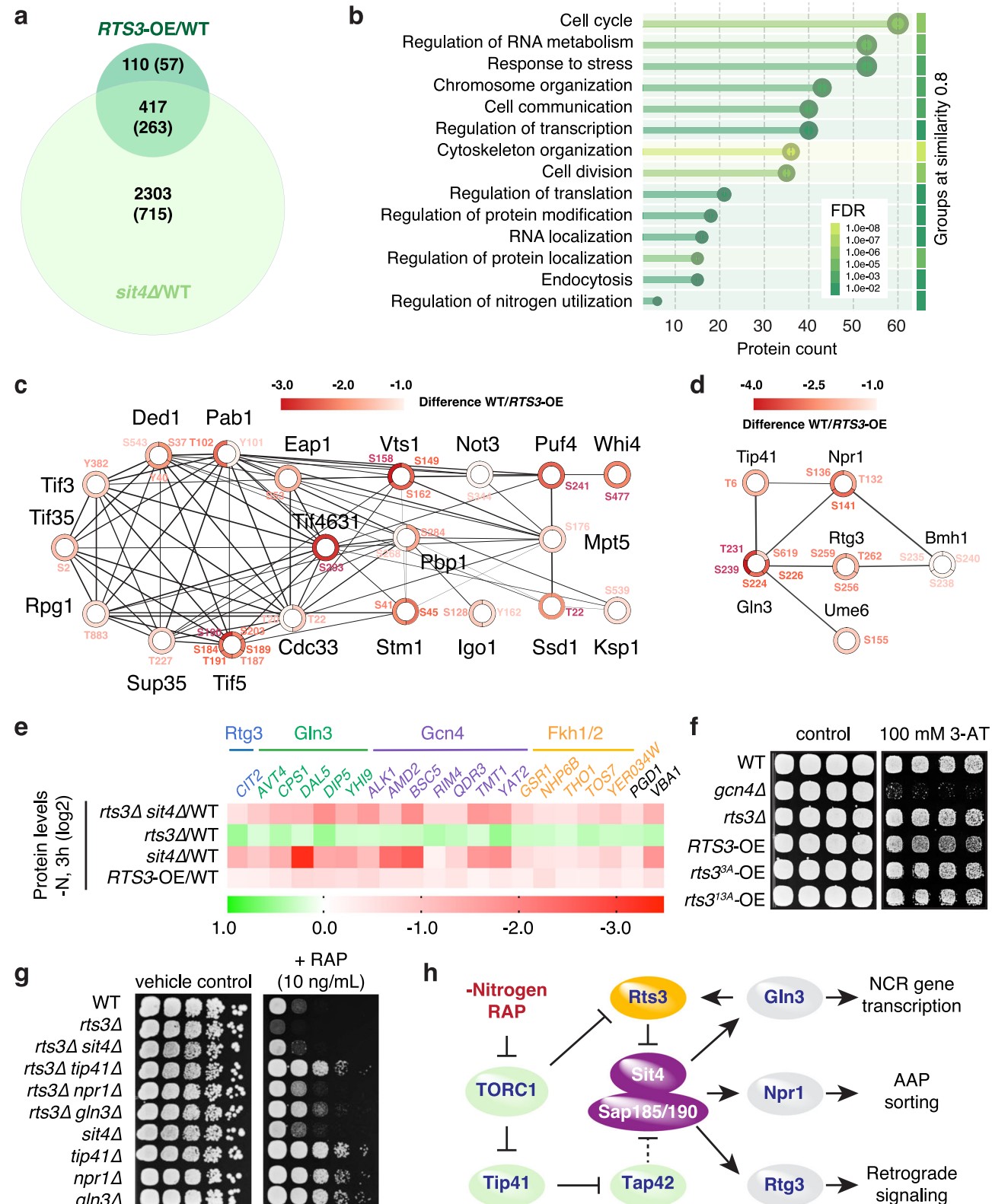

rapamycin sensitivity of *rts3Δ* cells (Fig. 5g). This genetic rescue confirms that Tip41-driven Sit4 activation and its downstream transcriptional (and Npr1-mediated nutrient transporter regulatory) outputs drive the stress vulnerability of these mutants. These data argue that Rts3 primarily affects the Sit4-Sap185/190 phosphatase, explaining why the Rts3 dosage alters stress phenotypes independently of Sch9 or Lst4 (Fig. 3c). Integrating these findings with our expression and turnover analysis reveals a feedback loop that temporally gates entry into and exit from quiescence (Fig. 5h). Under nitrogen limitation, TORC1 inhibition activates Sit4-Sap185/190 via the Tip41-Tap42 axis, resulting in dephosphorylation and activation of Gln3, Rtg3, and Npr1. In turn, Gln3/Gat1 drives *RTS3* transcription, and accumulating Rts3 protein binds Sit4 to limit downstream signaling. Upon nutrient restoration, TORC1 reactivation places Sit4 again under Tip41-Tap42[8] control and promotes SCF[Cdc4]-dependent clearance of Rts3, enabling exit from quiescence. This dual regulation − coupling transcriptional

**Fig. 5 | Rts3 constrains Sit4 to gate quiescence programs. a** Venn diagram of phosphosites and proteins (in brackets) upregulated (>2-fold, FDR < 0.05, $n = 5$ biological replicates) in *sit4Δ* (light green) and *RTS3*-OE (dark green) vs wild-type cells. Label-free phosphoproteomics was performed on lysates of rapamycin-treated (200 ng/mL; 30 min) WT, *sit4Δ*, and *RTS3*-OE (overexpressing GFP-Rts3 from *GPD* promoter) cells. **b** GO terms enriched among shared Sit4/Rts3-regulated phosphoproteins implicate translation and cell cycle control. **c, d** STRING-based networks of Rts3-regulated phosphoproteins (also Sit4-dependent) linked to nitrogen metabolism (**c**) and translation (**d**). Node color indicates fold-change of phosphosites; edge thickness indicates confidence of protein interaction (log2 values). **e** Heat map of proteins upregulated in a Sit4-dependent, Rts3-opposed manner during nitrogen starvation (3 h; FDR < 0.05, $n = 4$ biological replicates). Proteins were identified using label-free quantitative whole-proteome analysis with the indicated strains and grouped by transcription factor regulation (SGD annotation). **f** Rts3 overexpression sensitizes cells to 3-AT. Serial dilution assay of WT, *gcn4Δ*, *rts3Δ*, and *RTS3*-, *rts3^3A^*-, and *rts3^I3A^*-OE strains on SD + Arg/Trp plates ± 3-AT

(3-amino-1,2,4-triazole; 100 mM) incubated for 3 days at 30 °C. **g** Rapamycin sensitivity of *rts3Δ* is rescued by *SIT4*, *NPR1*, *GLN3*, or *TIP41* deletion. Spot assay (10-fold dilutions) of WT and mutant strains on YPD ± rapamycin (RAP; 10 ng/mL). **h** Model of Rts3 function in TORC1-Sit4 signaling. In growing cells, TORC1 promotes Rts3 phosphorylation and degradation. Upon nitrogen or amino acid (AA) deprivation, or rapamycin (RAP) treatment, TORC1 is inhibited, leading to Sit4 activation through Tip41 activation and Tap42 inactivation. Tip41-Tap42 association enhances Sit4 activity by displacing Tap42 from Sit4, thereby allowing free Sit4 to interact with Sit4-associated proteins (SAPs); consequently, Tap42 indirectly inhibits Sit4-Sap185/190 interaction (dashed bar). Activated Sit4 dephosphorylates and activates the transcription factors Gln3, Rtg3, and the kinase Npr1, promoting downstream processes such as nitrogen catabolite repression (NCR), retrograde response, and nutrient transporter regulation. Gln3 induces *RTS3* transcription, and Rts3 in turn inhibits Gln3 through Sit4 inhibition, forming a negative feedback loop. Arrows indicate positive regulatory interactions; bars denote inhibitory effects.

induction by GATA factors with rapid degradation via the TORC1-SCF^Cdc4^ axis — likely provides snap-switch kinetics for dynamically tuning quiescence and recovery.

## Discussion

Our work demonstrates that Rts3 functions as a starvation-induced inhibitor that engages the Sit4-Sap185/190 phosphatase module to couple translational restraint with transcriptional remodeling during nitrogen stress. Mechanistically, Rts3 positions an acidic α-helix deep into the catalytic cleft of Sit4-Sap185/190, dampening its activity to fine-tune TORC1 outputs across translation, NCR, retrograde signaling, and permease regulation. This mode of control fits seamlessly with a previous model in which Sit4-Sap185/190 complexes are indispensable for GAAC-mediated translational repression, Gln3/Gat1-dependent activation of NCR genes, and Rtg1/3-driven induction of *CIT2*[13].

Notably, loss of Sap185/190 phenocopies *RTS3* overexpression in conferring rapamycin resistance[14]. This, together with our observation that *rts3Δ* elevates NCR and retrograde programs, reinforces the idea that Rts3 exerts its primary effect through this branch. It functions as a molecular "dimmer switch" on TORC1-downstream programs. This rheostat function explains why loss of Rts3, which unleashes Sit4 activity, produces a kinetic acceleration of dephosphorylation that is more difficult to capture in static profiling than the blockade imposed by overexpression. Functionally, this may preserve a minimal level of active translation through different mechanisms to sustain a reversible quiescent state.

The Gln3-dependence of Rts3 expression implies a potential role during growth on poor nitrogen sources. Consistent with this, *RTS3*-overexpressing cells exhibited a mild growth defect on proline, whereas *rts3Δ* cells displayed a modest growth advantage (Supplementary Fig. 2e). This indicates that Rts3 functions as a rheostat to dampen signaling when nutrients are scarce, preventing excessive anabolic commitment. Such regulation is vital for the stability of the quiescent state, as loss of Rts3 significantly compromised chronological lifespan (Supplementary Fig. 2f and ref. [48]). At the cellular level, Rts3 associates with 80S ribosomes during starvation and limits Sit4-mediated dephosphorylation of the hibernation factor Stm1 (thereby preserving the TORC1-dependent mark) that balances ribosome homeostasis[44]. Importantly, the same small helix surface that mediates Sit4-Sap185/190 binding is required for biochemical activity and rapamycin resistance, tying physical association directly to functional output. Thus, Rts3 is not a generic adaptor but a precision partitioner of the Sit4 network, carving out a distinct signaling branch from broader Sit4 outputs. Furthermore, the phosphoproteomic overlap, where Rts3 targets appear as a strict subset of Sit4 targets rather than a unique signature, suggests that Rts3 does not redirect Sit4 to novel substrates or alter SAP selection. Instead, it selectively gates the activity of specific, pre-formed Sit4-SAP holoenzymes. Such

architectural compartmentalization allows targeted control of NCR/retrograde signaling and likely also translational programs while leaving other Sit4-associated processes untouched. This illustrates a general principle in phosphatase biology where specificity emerges from selective engagement of protein multimers and holo-complexes rather than global modulation.

Our data also reveal that Rts3 is subject to dual-mode regulation: transcriptional induction by the GATA factors Gln3 and Gat1 upon TORC1 inhibition (consistent with the robust enrichment of Rts3 in rapamycin-treated cells shown in Fig. 1b), and rapid posttranslational clearance via a TORC1-SCF^Cdc4^-proteasome axis during exponential growth and upon nutrient restoration. This architecture is well-suited to produce threshold-driven kinetics for entry into and exit from quiescence, coupling transcriptional remodeling to translational restraint in a temporally precise manner. Epistasis analyses support this wiring, showing that the rapamycin hypersensitivity of *rts3Δ* is suppressed by loss of Sit4 or its downstream effectors (Gln3, Tip41, and Npr1), consistent with Rts3 buffering excessive Sit4 activity during TORC1 inhibition.

From an evolutionary perspective, the yeast model suggests that similar inhibitory mechanisms may exist in higher eukaryotes. The human PP6 catalytic subunit (PP6C), orthologous to Sit4, has been implicated in colorectal cancer progression and glioblastoma radioresistance[49,50]. Our data thus raise the possibility that structured docking-motif inhibitors akin to Rts3 could modulate PP6 activity in human cells. Targeting this axis with synthetic mimetics may offer therapeutic potential in diseases characterized by dysregulated phosphatase signaling.

## Methods

### Yeast strains, plasmids, and growth conditions

*Saccharomyces cerevisiae* strains and plasmids are listed in Supplementary Table 1. Gene deletion was performed using the pFA6a system-based PCR-toolbox[51]. Gibson assembly was employed for generating plasmid constructs. Unless otherwise stated, yeast strains were grown to mid-log phase in synthetic dextrose (SD) medium (0.17% yeast nitrogen base, 0.5% ammonium sulfate, and 2% glucose) at 30 °C. For spot growth assays, rapamycin concentrations were titrated based on the media composition (rich YPD vs synthetic SD) to compensate for the increased sensitivity of yeast cells to TORC1 inhibition in synthetic media. To assess growth on proline, prototrophic yeast strains were diluted to an initial $OD_{600}$ of 0.02 in medium containing 0.1% proline, 0.17% yeast nitrogen base, and 2% glucose. Cultures were incubated in 100-well honeycomb plates at 30 °C, and growth was monitored using a Bioscreen C automated system (Thermo Labsystems). Optical density ($OD_{600}$) was recorded every 30 min for 60 h, with medium-amplitude shaking for 30 s prior to each measurement (paused 10 s before reading).

## Global PPase profiling

Wild-type *S. cerevisiae* cells were grown in YPD medium (1% yeast extract, 2% peptone, 2% glucose) at 30 °C to mid-log phase, treated or not with rapamycin (200 ng/mL) for 30 min, harvested by vacuum filtration, flash-frozen in liquid nitrogen, and stored at −80 °C until further processing. The experiment was performed in three biological replicates. For phosphatase inhibitor beads, 1.5 g of NHS-activated Sepharose beads were washed three times with 5 mL of 1 mM HCl, followed by 2 times 5 mL of coupling buffer (100 mM NaHCO$_3$, pH 8.2). 1 mg of aminoethanethiol-MC-LR was resuspended in 100 μL of methanol and coupled to the activated beads in coupling buffer for 1.5 h at room temperature. Cell lysates were incubated with 20 μL of beads on a rotor at 4 °C for 2 h. Beads were spun down and washed three times with phosphatase pulldown buffer[24,25]. Proteins bound to beads were digested with trypsin prior to LC-MS/MS analysis.

## Biotin proximity labeling and Rts3 proximome mapping

Yeast strains (MP7079 [*RTS3p-TurboID-V5-RTS3*] and LD7381 [*RTS3p-TurboID-V5-Envy*; negative control]) were grown to mid-log phase in 1 liter of synthetic dextrose complete (SDC) medium lacking uracil. Cells were treated with biotin (50 μM) for 1 hour and rapamycin (200 ng/mL) for 30 min. After treatment, cells were collected by vacuum filtration, washed with 200 mL of ice-cold 50 mM Tris.Cl (pH 7.5), flash-frozen in liquid nitrogen, and stored at −80 °C until further processing. Cells were disrupted using glass beads (0.25–0.50 mm diameter; Retsch) in 3 mL of RIPA buffer (50 mM Tris.Cl, pH 7.5; 150 mM NaCl; 1.5 mM MgCl$_2$; 0.1% SDS; 1% Triton X-100; 0.5% sodium deoxycholate; 1× cOmplete EDTA-free protease inhibitor cocktail [Roche]) using a Precellys homogenizer (6 × 30 s at 5000 rpm, 1 min pause between cycles). Lysates were clarified by centrifugation (1500 × *g*, 7 min, 4 °C), and total protein concentration was measured and adjusted using the Pierce™ BCA Protein Assay Kit (Thermo Fisher Scientific). NanoLINK® Streptavidin Magnetic Beads (150 μL; SoluLink) were added to the lysates and incubated at room temperature for 1 hour with constant rotation. Beads were subsequently washed five times with 100 mM ammonium bicarbonate and stored at −80 °C until further processing. The experiment was performed in three biological replicates.

Biotinylated proteins bound to streptavidin beads were extracted in 8 M urea in 100 mM ABC buffer and reduced with 1 mM DTT. The samples were loaded on the 10 kDa cutoff filter and spun at 8000 × *g* for 30 min. Proteins were alkylated using 5.5 mM iodoacetamide for 10 min at room temperature and washed twice with 100 mM ABC buffer. Finally, the proteins were digested on-beads with trypsin (Promega) overnight on a thermomixer at 550 rpm and 37 °C. Tryptic peptides were purified by STAGE tips, and LC-MS/MS measurements were performed by data-dependent analysis on a QExactive HF-X mass spectrometry coupled to an EasyLC 1200 nanoflow-HPLC (all Thermo Scientific). MaxQuant software (version 1.6.2.10)[52] was used for analyzing the MS raw files for peak detection, peptide quantification and identification using a Uniprot *S. cerevisiae* (2016) database. Carbamidomethylcysteine was set as fixed modification and oxidation of methionine was set as variable modification. The MS/MS tolerance was set to 20 ppm, and four missed cleavages were allowed for Trypsin/P as enzyme specificity. Based on a forward-reverse database, protein and peptide FDRs were set to 0.01, minimum peptide length was set to seven, and at least one unique peptide had to be identified. The match-between run option was set to 0.7 min. MaxQuant results were analyzed using Perseus software (version 1.6.2.3)[53].

Label free quantification measurements were obtained for all samples. Protein IDs with minimally two valid values out of three replicates in tested IP samples were considered for analysis. log2-transformed iBAQ values normalized to the median of the respective biological replicate were used for analysis. Missing values in control samples were imputed using a normal distribution with a width of 0.3 and down shift of 1.8. Statistical significance using Student's *t* test with

p values $p < 0.05$, and FDR corrected p values $q < 0.05$ shown in respective figures. Perseus software (version 1.6.2.3) was used for all statistical analyses.

## Label-free phosphoproteome analysis

Indicated yeast strains were grown in SDC medium lacking histidine to mid-log phase and treated with rapamycin (200 ng/mL) for 30 min. After addition of 6.5% trichloroacetic acid, cells were collected by centrifugation, and washed with ice-cold acetone. The resulting cell pellets were dried under vacuum and stored at −80 °C until further processing. The experiment was performed in five biological replicates.

MS sample preparation was performed as described in Hu et al.[54]. Briefly, proteins were extracted by disrupting yeast cells in 8 M urea using a bead beater and digested by Lys-C for 4 h at room temperature. Before overnight trypsin digestion, the concentration of urea was reduced to 1 M. The next day, peptides were purified using reversed phase cartridges (RP-S-5μl, Agilent) and eluted in 80% acetonitrile with 1% TFA for phosphopeptide enrichment using Fe(III)-NTA cartridges (Agilent)[55]. The flow-through was used for non-phosphopeptide analysis and PTM normalization.

LC-MS/MS measurements were performed on an Exploris 480 mass spectrometer coupled to an EasyLC 1200 nanoflow-HPLC (phospho peptides) or on a QExactive HFX mass spectrometer coupled to a Vanquish Neo UHPLC System (whole proteome, all Thermo Scientific). Peptides were separated on a fused silica HPLC-column tip (I.D. 75 μm, New Objective, self-packed with ReproSil-Pur 120 C18-AQ, 1.9 μm (Dr. Maisch) to a length of 20 cm) using a gradient of A (0.1% formic acid in water) and B (0.1% formic acid in 80% acetonitrile in water). Spray voltage was set to 2.3 kV and the ion-transfer tube temperature to 250 °C; no sheath and auxiliary gas were used. Mass spectrometers were operated in data independent mode; after each survey scan (mass range *m/z* = 400–1200; resolution: 120,000), 34 DIA scans with an isolation width of 24 *m/z* were performed covering a total range of precursors from 400 to 1200 *m/z*. AGC target value was set to 300%, resolution to 30,000 and normalized stepped collision energy to 25.5%, 27%, and 30%. The MS raw files were analyzed using Spectronaut software version 19 and directDIA+ workflow with standard settings using a Uniprot full-length yeast database and common contaminants such as keratins and enzymes used for digestion as reference. Statistical significance was determined using pairwise Student's *t* tests with a permutation-based FDR of <0.05. For biological network definition and Venn diagram overlap analysis (Fig. 5), an additional fold-change cutoff of >2 (log2 > 1) was applied.

Functional enrichment and interaction analyses were carried out using Cytoscape v3.10.3 (cytoscape.org) with the ClueGO plugin v2.5.3, STRING database[56], and Omics Visualizer[57]. GO enrichment for the cellular compartment category was assessed relative to the *S. cerevisiae* genome background, applying GO-term fusion to reduce redundancy.

## Comparative whole-proteome analysis

Indicated yeast strains were grown in SDC medium lacking uracil to mid-log phase and subsequently subjected to nitrogen starvation for 3 hours. Proteins were extracted from dry TCA-acetone-treated cell pellets (prepared as described above) using RIPA buffer supplemented with 0.17% ammonium hydroxide. Cell disruption was performed with glass beads in a Precellys homogenizer (3 × 90 seconds at 5000 rpm, 1 min pause between cycles). Protein concentration was determined using the BCA assay and adjusted to 1 mg/mL. Samples were treated with 0.5 μL Benzonase® Nuclease (Millipore) for 30 minutes on ice. The experiment was performed in four biological replicates.

MS sample preparation was performed using autoSP3 method described in Sanders et al.[58]. Briefly, proteins were reduced and alkylated using DTT and CAA and precipitated on Sera-Mag speedbeads

(GE Healthcare). After washing, proteins were digested overnight with trypsin.

LC-MS/MS measurements were performed on an Orbitrap Astral mass spectrometer coupled to a Vanquish Neo UHPLC System (all Thermo Scientific). Peptides were separated on a fused silica HPLC-column tip (I.D. 75 μm, New Objective, self-packed with ReproSil-Pur 120 C18-AQ, 1.9 μm (Dr. Maisch) to a length of 14 cm) using a gradient of A (0.1% formic acid in water) and B (0.1% formic acid in 80% acetonitrile in water). Spray voltage was set to 2 kV and the ion-transfer tube temperature to 280 °C; no sheath and auxiliary gas were used.

Mass spectrometer was operated in data-independent mode; 149 DIA scans with an isolation width of 4 $m/z$ covering a total range of precursors from 380-980 $m/z$ were performed on the Astral analyzer with an AGC target of 300%, 4 ms injection time and a normalized collision energy of 25%. The DIA loop was interrupted every 0.6 s by a survey scan using the Orbitrap detector (mass range $m/z = 380–980$; resolution: 240,000). The MS raw files were analyzed using Spectronaut software version 20 and directDIA+ workflow with modified standard settings (missed cleavage 3, all QValue and PEP cutoffs were set to 0.01) using a Uniprot full-length yeast database and common contaminants such as keratins and enzymes used for digestion as reference.

## Immunoblot analysis

Whole cell lysates were prepared as previously described[59]. Proteins were separated by SDS-PAGE and transferred onto nitrocellulose membranes. A complete list of primary and secondary antibodies, including working dilutions, is provided in Supplementary Table 1.

## Co-immunoprecipitations

For PP2A interaction assays, yeast cultures were grown exponentially in synthetic dextrose complete medium lacking ammonium sulfate (SDC-AS) and treated with 200 ng/mL rapamycin or vehicle for 30 min. Cells were collected by filtration and flash-frozen. Cell pellets were lysed by mechanical disruption with glass beads in a Precellys homogenizer in ice-cold lysis buffer (50 mM HEPES, pH 7.4; 150 mM NaCl; 10 mM MgCl$_2$) supplemented with 0.1% NP-40, protease- and phosphatase-inhibitors. Lysates were clarified by centrifugation and incubated with GFP-Trap® magnetic agarose (ChromoTek) for 2 h at 4 °C with rotation. Beads were then washed three times with lysis buffer, and bound proteins were eluted by the addition of SDS-PAGE sample buffer and heating at 95 °C for 5 min. Input and IP eluates were resolved by SDS-PAGE and analyzed by immunoblotting with the indicated antibodies. A pre-treatment of the membrane with 0.2 M NaOH served to demethylate PP2A catalytic subunits and allowed detection by immunoblotting.

To probe the interaction of Sit4 with SAPs, yeast strains LD8617, LD8616, and LD8593 were grown in YPD medium and shifted to SD medium lacking nitrogen, supplemented with rapamycin (200 ng/mL), for 30 min. Cultures (400 mL per strain) were harvested by centrifugation at 1000 × $g$ for 5 minutes at room temperature. Cells were aliquoted into 1.5 mL screw-cap tubes, flash-frozen in liquid nitrogen, and stored at −80 °C until further processing. Cells were lysed using a sequential bead-beating protocol in 1.2 mL lysis buffer containing 50 mM MES (pH 6.5), 150 mM NaCl, 1× cOmplete EDTA-free protease inhibitor cocktail (Roche), and 0.5% Triton X-100. Cell disruption was performed using a FastPrep-24 instrument (MP Biomedicals) for 45 s at power level 6.5. Lysates were clarified by centrifugation at 1500 × $g$ for 2 min at 4 °C. Protein concentration was determined using the BCA Protein Assay Kit (Thermo Fisher Scientific) and adjusted to equal levels before immunoprecipitation. Clarified lysates were incubated with anti-GFP mouse antibody (Roche; 1:700) overnight at 4 °C with constant rotation. In parallel, Protein G Dynabeads™ (Invitrogen) were blocked by overnight incubation with untagged yeast lysate at 4 °C. The following day, antibody-lysate mixtures were incubated with the blocked beads for 4 h at 4 °C. Beads were washed three times with lysis buffer, and bound proteins were eluted with Laemmli buffer (15 min at 65 °C). Eluates were analyzed by SDS-PAGE and western blotting as described above.

## Yeast two-hybrid assay

Protein interactions were analyzed using the split-ubiquitin DUAL-membrane System 3 (Dualsystems Biotech AG). Tco89 was cloned into the bait vector pCabWT, Rts3 into the prey vector pPR3-N. The bait-prey combinations were transformed into *S. cerevisiae* NMY51, using positive (pAI-Alg5) and negative (pDL2-Alg5) control vectors in parallel. Transformants were spotted as 10-fold serial dilutions onto SD-Leu/Trp, SD-Leu/Trp/His, and SD-Leu/Trp/Ade plates, supplemented (where indicated) with 75 μM MG-132, 0.5% DMSO, and 0.003% SDS. Plates were incubated for 3 days at 30 °C. Protein-protein interactions were scored as growth under selective (-His or -Ade) conditions.

## RNA isolation and RT-qPCR analysis

Yeast cells were collected at an OD of 0.6–0.8 (2 mL culture aliquot) by centrifugation at 2500 × $g$ for 2 min. Cell disruption was performed with glass beads in a Precellys homogenizer (3 × 30 seconds at 5000 rpm, 1 min pause between cycles), and RNA was subsequently extracted from the lysate using the Quick-RNA kit (Zymo Research) according to the manufacturer's instructions. *RTS3* transcript levels relative to *TBP1* reference gene were analyzed using Luna® Universal One-Step RT-qPCR Kit (New England Biolabs). 5 ng of total RNA were added to the 20 μL reaction mix, the final concentration of each primer (listed in Supplementary Table 1) was 0.4 μM. Reactions were performed in triplicate. PCR cycle conditions consisted of 10 min of reverse transcription (55 °C), 1 min of initial denaturation (95 °C), followed by 40 cycles of 10 s at 95 °C and 30 s at 60 °C. Green fluorescence was monitored after each extension step. The amount of the respective transcript was determined for 3 biological replicates using the ΔΔCt method[59].

## Fluorescence microscopy

Yeast cells were grown in SDC-AS medium to log phase at 30 °C. Nitrogen starvation and amino acid refeeding were induced as described[60]. Cells were imaged using an inverted spinning-disk confocal microscope (Nikon Ti-E; VisiScope CSU-W1) equipped with a dual-camera system (Hamamatsu Orca Quest C15550-20UP) and a 100× NA 1.3 oil-immersion Nikon CFI objective. Z-stacks of eleven images at 0.2 μm spacing were acquired.

## Computational modeling and MDs simulations

A structural model of the Sit4-Sap185-Rts3 trimeric complex was generated using AlphaFold-Multimer v2 (https://doi.org/10.1101/2021.10.04.463034). Among five predicted models, the one with the highest combined model confidence score (ipTM + pTM = 0.7) was selected for further analysis. Predicted disordered regions in Rts3 and Sap185 were excluded prior to MD simulations. The final construct used for simulation included residues 52–123 of Rts3, residues 16–796 of Sap185, and the full-length Sit4. The system was solvated with the TIP3P water model and neutralized by adding Na$^+$ and Cl$^-$ ions to a final concentration of 0.15 M. Energy minimization was carried out using the steepest descent algorithm. An initial 2 ns NVT equilibration was performed at 300 K using the v-rescale thermostat with a coupling constant $\tau_T = 1$ ps[61]. This was followed by a 2 ns NPT equilibration at 310 K using the same thermostat and the Berendsen barostat with $\tau_P = 1$ ps[62]. The system was subsequently subjected to a 320 ns production MD simulation in an NPT ensemble at 310 K. Temperature was controlled using the v-rescale thermostat, and pressure was maintained with the Parrinello-Rahman barostat[63], both with $\tau_T$ and $\tau_P = 1$ ps. Electrostatic interactions were calculated using the Particle Mesh Ewald (PME) method[64] with a real-space cutoff of 1.2 nm. A 1.2 nm cutoff was also

applied to van der Waals interactions. All simulations were carried out using the CHARMM36 force field[65] and GROMACS v2021.5 software (https://doi.org/10.5281/zenodo.4457591). Hydrogen bonds were identified using the gmx hbond module in GROMACS v2021.5. Electrostatic potentials of Sit4 were computed using APBS v3.4.1[66] and visualized in ChimeraX[67].

### Sit4 in vitro PPase assay

Purification of HA-tagged Sit4 from *S. cerevisiae* and the subsequent in vitro PPase assay using the KRpTIRR phosphopeptide (MedChemExpress) as a substrate were performed as described[68]. Briefly, HA-Sit4, immunoprecipitated from yeast cells and activated with 5 mM ascorbate, was pre-incubated on ice for 20 min with the indicated variants of recombinant $His_6$-tagged Rts3. These were expressed in *E. coli* Rosetta cells, purified using Ni-NTA agarose (Qiagen) in phosphate-free buffers according to the manufacturer's instructions, and buffer-exchanged into TBS (Tris-buffered saline) using PD-10 columns (Cytiva). PPase reactions were carried out for 3 hours at 30 °C in a thermoblock shaker at 600 rpm. Free phosphate release was quantified using the Biomol Green reagent (Enzo Life Sciences), and $IC_{50}$ values were determined by nonlinear regression using GraphPad Prism 10.

### TORC1 purification and in vitro kinase assay

TORC1 purification and in vitro kinase assay were performed as previously described[69]. Briefly, 3 μL of TORC1 purified from yeast cells were incubated with 1 μg of recombinant $His_6$-Rts3 in kinase buffer (50 mM HEPES, pH 7.5; 400 mM NaCl; 8.75 mM $MgCl_2$ or 3.5 mM $MnCl_2$; 7 μCi of $[\gamma\text{-}^{32}P]$-ATP [6000 Ci/mmol]; and 315 μM cold ATP) for 30 min at 30 °C. The reaction (final volume 20 μL) was stopped by adding 4 μL 6× Laemmli SDS sample buffer and denaturing for 5 min at 65 °C. The samples were resolved by SDS-PAGE and analyzed by SYPRO Ruby staining and autoradiography. Wortmannin was used to inhibit TORC1 kinase activity as described[70], since rapamycin inhibition relies on the FKBP12 cofactor, which is absent in the purified fraction.

### Purification of cytoplasmic ribosomes

To purify mature, cytoplasmic ribosomes, we adapted a previously established pulse-labeling protocol that selectively biotinylates AVI-tagged ribosomal proteins[45]. Accordingly, Rps2-TEV-HA-AVI was pulse-labeled with biotin in yIK127 cells. Cells were grown in SDC medium with low biotin levels (0.5 nM). At $OD_{600}$ = 0.4, BirA-AID expression was induced by adding 5 nM estradiol. After 2 h, estradiol was washed out, and the pulse-labeling of Rps2 was initiated by supplementing the culture with 10 nM biotin and incubation for 1 h. At the same time, BirA-AID expression was terminated by the addition of indole-3-acetic acid (IAA; 1 mM). Both induced and control cultures were then washed twice with water, starved for nitrogen for 72 h at 30 °C, treated with cycloheximide (100 μg/mL, 10 min, 23 °C), harvested by centrifugation (1000 × *g*, 5 min, 4 °C), and snap frozen in liquid nitrogen.

For streptavidin pull-down, cell pellets were resuspended in lysis buffer (50 mM Tris-Cl, pH 7.5; 100 mM NaCl; 1 mM EDTA, pH 8.0; 20 mM $MgCl_2$; 100 μg/mL cycloheximide; 1 mM PMSF; 1× cOmplete™ EDTA-free protease inhibitor; 0.1% NP-40). Cells were disrupted with Zirconia/Silica beads (0.5 mm; Biospec) by vortexing (15 min, 2850 rpm, 4 °C). Lysates were clarified (20,000 × *g*, 20 min, 4 °C), and protein concentrations were determined by BCA assay and normalized across samples, which were then incubated with Dynabeads® MyOne™ Streptavidin C1 (Invitrogen) for 1 hour at 4 °C. After washing the beads twice with lysis buffer, PBS (phosphate-buffered saline) containing 0.1% NP-40, and PBS without detergent, the bound proteins were eluted in urea containing SDS-PAGE sample buffer (65 °C, 15 min), resolved by SDS-PAGE, and analyzed by immunoblotting with the indicated antibodies. Three independent experiments were performed; a representative blot is shown in Supplementary Fig. 2b.

### Polysome profiling

Indicated strains were grown in SDC medium lacking uracil until reaching an $OD_{600}$ of 0.6, washed twice with water, and starved for nitrogen for 72 h at 30 °C. Polysome profiling was performed as described previously[44], with slight modifications. In brief, cells were treated with cycloheximide (100 μg/mL) for 10 min at 23 °C, pelleted by centrifugation (5 min, 1000 × *g*), washed once with ice-cold PBS and once with lysis buffer (20 mM Tris-Cl, pH 7.5; 150 mM KCl; 5 mM $MgCl_2$; 1% Triton X-100; 1 mM DTT; 100 μg/mL cycloheximide; 0.1 mM PMSF), and then resuspended in 800 μL lysis buffer. Cells were disrupted as described above for the purification of cytoplasmic ribosomes. A total of 500 μL extract was loaded onto a 10–50% sucrose gradient prepared in lysis buffer without Triton X-100 and centrifuged at 39,000 rpm for 2 h at 4 °C using an SW 41 Ti rotor (Beckman Coulter). Subsequently, the gradient was fractionated with the Triax™ Flow Cell (Biocomp Instruments), and absorbance was recorded at 260 nm. Four independent experiments were performed. An unpaired Student's *t* test was used for statistical analysis.

### Chronological life span assay

For chronological lifespan analysis (survival assays), cells were grown in SDC medium to stationary phase (72 hours) and maintained in the spent medium. At indicated time points, aliquots were removed, diluted, and plated on YPD agar to determine colony-forming units (CFUs). Survival is expressed as a percentage of CFUs relative to the first day of the stationary phase (Day 0).

### Statistical analysis

Statistical parameters are reported in the figure, supplementary figure, and supplementary data legends, and the corresponding methods details.

### Reporting summary

Further information on research design is available in the Nature Portfolio Reporting Summary linked to this article.

## Data availability

All the data supporting the results of this study are presented within the article, the Supplementary Information and the Supplementary Files. Source data for gel images, microscopy images, and graphs can be found in Mendeley Data (https://data.mendeley.com/preview/pj2dkx8jw3?a=252ec8da-f7b8-4e74-be44-39b90ce1a242). Proteomics data are freely available via the PRIDE repository (https://www.ebi.ac.uk/pride/), data identifier PXD069824. Additional information can be obtained from the corresponding author. Source data are provided with this paper.

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

## Acknowledgements

We thank Susanne Stumpe and Janny Tilma for technical assistance, and Jiří Veis and Egon Ogris for strains and methodological advice. This work was supported by the Canton of Fribourg (to C.D.V. and J.D.), the Swiss National Science Foundation (184671/214824 to C.D.V.; 212187 to JD, 229588 to C.D.V. and J.D., and 200426 and 236080 to M.P.). Insa Klemt was financed by an EMBO post-doctoral fellowship (ALTF 475-2024) and supported by ETH Zürich.

## Author contributions

Conceptualization: L.D. and C.D.V.; methodology: L.D., M.-P.P.-G., I.K., M.J., J.A., M.S., C.A.J., R.L.C., D.S., and Z.H.; formal analysis: L.D., I.K., J.D., M.P., and C.D.V.; writing–original draft: L.D., I.K., J.A., J.D., and C.D.V.; writing–review & editing: L.D., M.-P.P.-G., J.D., and C.D.V.; visualization: L.D., J.A., M.-P.P.-G., I.K., M.J., J.D., and C.D.V.; funding acquisition: M.P., J.D., and C.D.V.

## Competing interests
The authors declare no competing interests.

## Declaration of AI-assisted technologies
During the preparation of this manuscript, the authors used GPT-5 via Microsoft Copilot and Gemini 3 to enhance the clarity and conciseness of the text. The tool was applied under human oversight, and the authors carefully reviewed and edited the output. The authors take full responsibility for the content of the published article.
