## [Transparent Peer Review file · Nature Communications]

The nitrogen starvation induced inhibitor Rts3 restrains Sit4/PP6 to gate quiescence downstream of TORC1

Corresponding Author: Professor Claudio De Virgilio

Version 0:

Reviewer comments:

Reviewer #1

(Remarks to the Author)

Dokladal et al. describe a new feedback loop in the TORC1 signaling pathway, in which the starvation-induced protein Rts3 binds and inhibits the Sit4–Sap185/190 phosphatase complex, thereby dampening the dephosphorylation of targets involved in nitrogen metabolism and translation.

This is an important and technically impressive study that fills a major gap in our understanding of how TORC1 regulates phosphatase activity—a long-standing challenge in the field due to the technical difficulty of dissecting these interactions and phosphatase function. The authors use an elegant combination of biochemical assays, quantitative proteomics, and structural modeling to define Rts3's function with high precision.

The work is also clear and well written, and will make a valuable contribution to the TORC1 signaling field. The manuscript's main strengths are its rigor, depth of molecular characterization, and mechanistic insight. However, there are two areas for improvement that should be addressed to increase the paper's overall impact before publication.

1. Broader Function and Physiological Role of the Rts3 Feedback Loop

While the study convincingly defines the Rts3-dependent feedback mechanism and its biochemical connection to TORC1, the physiological significance of this feedback remains underexplored. The authors link Rts3 to stabilization of the quiescent state, but the impact on quiescence exit appears modest.

Because Rts3 is induced by GATA factors (Gat1/Gln3), it is likely expressed under nutrient-limiting conditions as well as during starvation. This raises the possibility that Rts3 serves to stabilize or attenuate TORC1 signaling in cells growing under poor nutrient conditions, not just in fully starved cells. Addressing this hypothesis—through discussion or additional experiments—would substantially strengthen the paper. For example, the authors could assess the growth or survival of *rts3Δ* cells in proline or other poor nitrogen sources could help clarify its physiological role. It would also be interesting to follow up on the argument that Rts3 stabilizes the quiescent state by looking at cell survival over long time periods of starvation in wt and *rts3Δ* cells.

2. Analysis and Interpretation of Rts3's Impact on Sit4–Sap185 Function

The authors demonstrate that Rts3 inhibits Sit4 activity *in vitro* and *in vivo*. However, the core analysis relies heavily on overexpression data in rapamycin-treated cells, where Rts3 is already induced and likely occupies most Sit4–Sap185 binding sites. In contrast, the impact of Rts3 deletion is underexplored.

A more detailed presentation and discussion of the *rts3Δ* proteomic data would clarify the true functional scope of Rts3. Specifically:

- Are the changes in phosphorylation upon deletion consistent with simple inhibition, or could Rts3 alter phosphatase specificity or Sit4 binding partner selection?
- How do the Rts3-dependent sites not found in the Sit4 delete fit with its mechanism of action?
- How was statistical significance defined (e.g., FDR correction, cutoffs)? Listing these thresholds would improve clarity and reproducibility.

Additionally, the text indicates that Rts3 binds the Sit4–Sap185 complex in a way that would inhibit its activity, but this is not clearly explained. The authors should specify where the Rts3 helix binds, how this position might block or modify access to substrates, and whether the model predicts simple inhibition versus altered specificity.

3. Specific Comments and Minor Revisions

- Line 48:

Should read “A subset of these enzymes interacts with, and are regulated by...”

- Line 90 and Table 1:

The statement that Rts3 was “significantly enriched” in rapamycin-treated cells needs clarification. The figure legend just restates this fact—please specify how enrichment was determined, including criteria or statistical thresholds used (e.g., fold-change, FDR).

- Promoter analysis section (Figure 2B–C):

The phrase “large-scale promoter annotations and motif analyses (Saccharomyces Genome Database)” is vague. Please clarify which studies or datasets were used and cite the primary literature rather than the database itself.

It is also unclear that Gat1 and Gln3 are the primary drivers of RTS3 expression. Figure 2C suggests additional regulators may contribute, as the induction is only partially blunted in the double mutant. Replotting the data (and that of Fig. 2b) on a log₂ scale could help visualize dynamic range and clarify this point.

- TurboID dataset:

With 764 significant interactors, the current analysis is limited. A more global interpretation—e.g., enrichment of functional categories or pathways—would help contextualize Rts3’s role beyond the specific Sit4-Sap185 and ribosomal interactions highlighted.

- Structural model and transition (text around Fig. 4):

The rationale for testing Rts3 as an inhibitor is not clear from the preceding description. Please explain how the predicted Rts3-binding site suggests inhibition of Sit4 activity.

- Figure 5 and phosphoproteomic analysis:

As discussed above, the logic of using Rts3 overexpression in rapamycin as the main readout is problematic, since Rts3 is already induced under these conditions. A deletion versus wild-type comparison in rapamycin would provide a cleaner view of Rts3 targets and should be emphasized in the analysis and discussion (referenced in Fig. 5E but not adequately integrated earlier).

- Physiological relevance:

The authors briefly mention a small delay in recovery from quiescence in *rts3Δ* cells, but this the physiological impact of the Rts3 feedback loop is not analyzed in depth. Further discussion or data addressing nutrient-limiting growth conditions (e.g., poor nitrogen sources) could significantly enhance the impact of the work.

Reviewer #2

(Remarks to the Author)

The study “The nitrogen starvation-induced inhibitor Rts3 restrains Sit4/PP6 to gate quiescence downstream of TORC1” by Dokládál et al. describes a novel mechanism by which TORC1 governs the balance between cell proliferation and entry into the quiescence programme in budding yeast. TORC1 regulates the activity of protein phosphatases such as Sit4 (PP6 in mammalian cells), thereby controlling Sit4-driven translational and transcriptional programmes.

In this study, the authors investigate the phosphatase landscape downstream of TORC1 by applying an unbiased phosphatase inhibitor beads-mass spectrometry approach to cells treated with rapamycin, thereby specifically inactivating TORC1. This strategy identifies Rts3 as the only phosphatase-associated factor consistently enriched upon TORC1 inhibition. The authors show that Rts3 contributes to the control of cellular quiescence. Its stability increases during nitrogen starvation, whereas nutrient re-addition triggers SCF-Cdc4-dependent ubiquitination and proteasomal degradation. Functionally, Rts3 reduces Sit4 activity during starvation. The work further reveals that Gln3, activated by Sit4, induces RTS3 expression, establishing a negative feedback loop that maintains quiescence at an appropriate depth and ensures that cells can resume growth once nutrients are restored.

Overall, the mechanism described is certainly interesting and novel, and the evidence presented by the authors supports the conclusions reached. The work addresses a relevant question in the field, and the findings are of significance to both the TORC1 and quiescence communities. The experimental design is sound, the methodology meets the expected standards in the field, and the PIB–MS approach is applied in a rigorous and unbiased manner.

Major Comments

1. Rts3 band patterns differ between the Figures 2A and 2D, especially at +AA 10 and +AA 60 minutes. Clarification or use of matched blots would improve consistency.
2. The slower-migrating forms of Rts3 appears only in *rpn5-1* (and with MG-132) but not in *pup1-1*. In addition, the +AA 60 band in *pup1-1* resembles wt. This inconsistency should be addressed.
3. Including TOR1-1 cells in Figure 2E would demonstrate that the mobility shift is TORC1-dependent rather than an artefact of rapamycin exposure.
4. Ubiquitinated-protein blots differ between Figures 2D and 2F and should be more comparable.
5. More methodological detail on the kinase assay should be added in the main text and figure legend, and the authors may wish to comment on the reason why they used wortmannin instead of rapamycin as in the rest of the manuscript for the same

purpose.

6. Including described Rts3 interactors as control in Figure 3A would confirm that the absence of interaction with Tpd3 and Pph21/22 is not due to low Rts3 abundance.

7. The experimental procedure for Figure 4C requires further explanation. The Rts3-3A mutant shows differential loss of interaction with Sit4, Sap185 (clear decrease) compared with Sap190 (not so clear decrease); this difference should be addressed and clarified.

8. The manuscript uses 4, 5 and 10 ng/ml rapamycin in different assays. A brief justification would help. Growth differences between Figure 4J (5 ng/ml) and Figure 5G (10 ng/ml) should be clarified. In Figure 4J first spot in rapamycin plate is growing weakly at 5ng/ml but in Figure 5G at 10ng/ml authors present better growth.

9. The phenotype observed on 3-AT plates in Figure 5F is quite subtle. Including a control strain to verify that the 3-AT assay is working effectively would support the conclusion. The authors may also wish to adjust the wording accordingly.

10. The genetic logic may not be intuitive to general readers. A more detailed explanation will benefit the manuscript. Authors could consider to use an additional experimental approach to show the same, which would strengthen the conclusion.

11. In the Discussion, the statement that Rts3 “counteracts TORC1-dependent phosphorylation of the hibernation factor Stm1 that balances ribosome homeostasis” implies enhanced Stm1 dephosphorylation, which seems at odds with the model in which Rts3 inhibits Sit4, the phosphatase expected to act on Stm1. This apparent contradiction should be addressed, unless this reviewer misunderstood.

Minor Comments

1. Strain names should be included in the figure legends to improve clarity of the experimental details.

2. As Figure 1B supports the proposed negative feedback loop, explicitly referring back to it when this mechanism is discussed later in the text would help the reader.

3. Page 4, line 113 onwards: a more detailed explanation would help readers follow Figure 1C. In addition, it may be worth reconsidering whether the term 'genetic analysis' is the most appropriate description in this context.

4. Figure 2C: as residual RTS3 expression persists in *gat1 gln3* deleted cells, the authors may wish to discuss additional pathways that could contribute to RTS3 expression.

5. Figure 3B: showing Rts3 levels expressed from the endogenous promoter and from ADH1 promoter could help define the expression threshold above which cells become rapamycin-insensitive.

6. Figure 3D: larger crops of cells would improve visibility of the phenotypes shown.

7. Figure 4E: some residues are difficult to see; alternative colouring may enhance clarity.

8. Page 34, line 897 (Figure 4): please clarify why “not shown”.

9. Page 34, line 912 (Figure 4): clarify whether expression was tested in asynchronously growing WT cells.

10. Page 8, line 211: additional detail on the phosphatase assay should be added to the main text.

Editorial Points

Page 49: rephrase “Tip1-mediated Tap42 sequestering”.

Line 229: “2720” rather than “2’720”.

Line 236: clarify the meaning of “network-level dissection”.

Line 242: specify the protein in which Ser41 and Ser45 are affected.

Line 254: clarify “shared Sit4- and Rts3-regulated nodes” for readers unfamiliar with STRING.

Line 271: clarify “permease regulatory”.

Line 280: rephrase the “dual-mode regulation–feedforward induction” sentence.

Line 807: correct “declaration od”.

Reviewer #3

(Remarks to the Author)

Reviewer #4

(Remarks to the Author)

In the manuscript by Dokládál et al., the authors investigate how protein phosphatases downstream of TORC1 are regulated by utilizing an approach that combines PPase inhibitor beads with mass spectrometry. They identify Rts3 as the PPase-associated protein markedly enriched upon rapamycin treatment. In addition, they demonstrate that Rts3 accumulates during nitrogen starvation and is degraded upon nitrogen readdition through TORC1- and SCF-dependent mechanisms. Rather than binding the canonical PP2A complex, Rts3 is suggested to act as a direct inhibitor of Sit4 by interacting with its catalytic cleft in complexes containing Sap185/190. Through this inhibition, Rts3 is proposed to restrain Sit4 activity during starvation, limit Gln3- and Rtg3-dependent transcription, and thereby fine-tune quiescence depth and promote efficient reactivation upon nutrient replenishment. Overall, most of the data are of high quality and seem to support the authors' conclusions. On the other hand, many of the experiments rely on overexpression systems, and thus, it is difficult to exclude the possibility that some of the observed effects may arise from overexpression artifacts.

Major points

- 1) The authors claim that Rts3 is a direct substrate of TORC1 by performing an in vitro kinase assay (Fig. 2G). Have the authors confirmed the physical interaction between TORC1 and Rts3, which would further support their claim?
- 2) In Fig. 3B, the authors shows that the overexpression of Rts3 from ATD promoter confers rapamycin resistance on budding yeast cells. Since this experiment relies on the Rts3 overexpression, the observed rapamycin resistance might not reflect Rts3-mediated inhibition of Sit4, but could instead be an artifact of overexpression. To rule out this possibility, the same experiment should be performed using sit4 deletion mutant cells, in which rapamycin resistance would not be expected.
- 3) The authors validated the interaction model of Sit4 and Rts3 obtained from MD simulations by co-immunoprecipitation assays with alanine mutants of Rts3 (Fig. 4C, E and F). On the other hand, the importance of Sit4 residues R266, Y263, and Y124, which have been proposed to interact with Rts3, has not been experimentally validated. It would be valuable to assess the significance of these residues using co-immunoprecipitation (co-IP) with alanine-substituted Sit4 mutants.
- 4) In Fig. S1C, the growth delay observed in rts3Δ and GFP-RTS3-OE cells may not necessarily reflect a slower recovery from nitrogen starvation compared to wild-type cells, but could instead be due to an overall slower growth rate. This possibility could be addressed by examining growth curves using cells in logarithmic phase.
- 5) In the polysome analysis shown in Fig. S1D, no statistically significant difference was observed between wild-type and rts3Δ cells. Moreover, although the difference between wild-type and GFP-Rts3-OE cells is statistically significant, it is rather small, as noted by the authors. Nevertheless, as shown in Fig. S1C, both rts3Δ and GFP-RTS3-OE cells exhibit a similar delay in recovery from nitrogen starvation compared to wild-type cells. Therefore, I think that the authors' conclusion 'Together, these results confirm that Rts3 associates and constrains Sit4 on hibernating ribosomes, allowing efficient recovery from prolonged starvation, thus emphasizing its physiological relevance (lines 249-251)' is not supported by the data.

Minor points

- 6) In the input samples of Fig. 4C, a strong non-specific band is observed in addition to GFP-Rts3, whereas no such band appears in Fig. 4A. What accounts for this discrepancy?
- 7) A part of the immunoblotting image in Fig. 4C (right side) appears to be missing.
- 8) In Fig. 4E and F, the red numbering of the amino acid residues overlaps with the structure colored in magenta, making it difficult to distinguish.
- 9) Statistical analysis should be performed for Fig. 4H.
- 10) In line 125, 'In' should not be capitalized.

Version 1:

Reviewer comments:

Reviewer #1

(Remarks to the Author)

The authors did an excellent job of revising the manuscript to address all of the reviewers comments. I am happy with the paper, and have no more major concerns, but suggest that the authors spend a bit more time editing the discussion as some of the english still reads like a draft. For example, three sentences in lines 364 to 372 start with the word crucially, and the flow in this section could be improved (sentence to sentence linkages etc).

Reviewer #2

(Remarks to the Author)

I have reviewed the revised version of the manuscript. The authors have adequately addressed the comments and suggestions raised in the previous review, and the revisions have improved the overall clarity and coherence of the work.

The manuscript is now methodologically sound and clearly presented. I have no further major concerns and consider it suitable for publication in its current form.

In addition, I would like to note two minor typographical errors in the Supplementary section that should be corrected before publication. In both cases, the word Supplementary is misspelled (missing the letter "n"):

"Supplementary information contains Supplemetary Figures 1–3 and Supplementary Table 1 and can be found online at <https://nn>."

"Supplementary files contain Supplemetary Data 1–4 and can be found online at <https://nn>."

I would kindly ask the authors to correct these minor typographical errors to ensure consistency and accuracy in the Supplementary materials.

Reviewer #3

(Remarks to the Author)

Reviewer #4

(Remarks to the Author)

The authors have appropriately addressed my comments and concerns in the revised version of the manuscript. The paper has been substantially improved, and I have no further major issues to raise. In my view, the manuscript is now suitable for publication.

Responses to reviewers:

Reviewer #1

Major Comments

1. **Broader Function and Physiological Role of the Rts3 Feedback Loop.** While the study convincingly defines the Rts3-dependent feedback mechanism and its biochemical connection to TORC1, the physiological significance of this feedback remains underexplored. The authors link Rts3 to stabilization of the quiescent state, but the impact on quiescence exit appears modest. Because Rts3 is induced by GATA factors (Gat1/Gln3), it is likely expressed under nutrient-limiting conditions as well as during starvation. This raises the possibility that Rts3 serves to stabilize or attenuate TORC1 signaling in cells growing under poor nutrient conditions, not just in fully starved cells. Addressing this hypothesis—through discussion or additional experiments—would substantially strengthen the paper. For example, the authors could assess the growth or survival of *rts3Δ* cells in proline or other poor nitrogen sources could help clarify its physiological role. It would also be interesting to follow up on the argument that Rts3 stabilizes the quiescent state by looking at cell survival over long time periods of starvation in wt and *rts3Δ* cells.

We thank the reviewer for this insightful suggestion regarding the physiological scope of the Rts3 feedback loop. We agree that the GATA-dependent induction of Rts3 suggests it functions to fine-tune TORC1 signaling during growth on poor nitrogen sources, in addition to its role in starvation.

To address this, and to **replace** the less robust regrowth data (as discussed in the response regarding original Fig. S1c below), we performed two new sets of experiments that significantly strengthened our model:

1. **Growth on poor nitrogen source:** We assessed the growth of *rts3Δ* and *RTS3*-overexpressing (*RTS3*-OE) strains on proline, a poor nitrogen source where uptake is mediated by the Gln3-dependent permease Put4. Consistent with the reviewer's hypothesis that Rts3 attenuates signaling under these conditions, *RTS3*-OE cells displayed a mild growth defect on proline, likely due to excessive inhibition of Sit4 and subsequent repression of Gln3-dependent Put4 expression. Conversely, *rts3Δ* cells exhibited a modest growth advantage, consistent with unrestrained Gln3 activity enhancing proline uptake. These data are now included in the **new Figure S2e**.
2. **Chronological lifespan (CLS) analysis:** We assessed cell survival over long-term starvation. As hypothesized, *rts3Δ* cells displayed significant sensitivity to prolonged nitrogen starvation compared to the wild type. This confirms that Rts3 is critical for the stability and maintenance of the quiescent state. We have added these results to the **new Figure S2f** and the **Discussion**.

These combined results clarify the physiological role of Rts3: it acts as a rheostat to fine-tune growth on poor nitrogen sources and as a critical "brake" to ensure long-term survival during deep quiescence, rather than regulating the kinetics of quiescence exit. We have **expanded the Discussion and Results** sections to incorporate these findings.

2. **Analysis and Interpretation of Rts3's Impact on Sit4–Sap185 Function.** The authors demonstrate that Rts3 inhibits Sit4 activity *in vitro* and *in vivo*. However, the core analysis relies heavily on overexpression data in rapamycin-treated cells, where Rts3 is already induced and likely occupies most Sit4–Sap185 binding sites. In contrast, the impact of Rts3 deletion is underexplored. A more detailed presentation and discussion of the *rts3Δ* proteomic data would clarify the true functional scope of Rts3.

To address the functional scope of Rts3 more comprehensively, we performed a global label-free phosphoproteomic analysis of the *rts3Δ* mutant in parallel with the overexpression and *sit4Δ* conditions. Upon analyzing the *rts3Δ* dataset using the same stringent statistical criteria applied to the overexpression dataset (FDR < 0.05, Log2 FC > 1), we observed very few significant changes.

We attribute this outcome to a fundamental detection asymmetry inherent to the experimental conditions:

- (i) Overexpression (high signal-to-noise): Blocking Sit4 (*RTS3*-OE) leads to a massive accumulation of phosphorylation compared to the rapamycin-treated control (where Sit4 is active), resulting in a robust "gain of signal."
- (ii) Deletion (floor effect): In the *rts3Δ* mutant, Sit4 becomes hyperactive. However, since Sit4 is already active in the rapamycin-treated control, the dynamic range for further dephosphorylation is compressed. This creates a "floor effect" where the loss of residual phosphate groups is often indistinguishable from the limit of detection at a global statistical level.

Importantly, despite these statistical limitations, we inspected biologically relevant targets from our network. We found that phosphorylation sites on the ribosome hibernation factor Stm1 (Ser⁴¹ and Ser⁴⁵) displayed clear trends in the expected direction (reduced phosphorylation consistent with Sit4 hyperactivity), supporting our model.

To reflect this analysis, we have **added the *rts3Δ* dataset to Supplementary Data 3** and updated the manuscript in three key areas. In the **Results**, we now explicitly report the *rts3Δ* profiling alongside the overexpression data, describing the "floor effect" to explain the asymmetry in global hits. We further refined the translation subsection to note that while *RTS3* overexpression prevents Stm1 dephosphorylation, the *rts3Δ* mutant displays a trend toward reduced phosphorylation at Ser⁴¹/Ser⁴⁵, consistent with unrestrained Sit4 activity. Finally, we **revised the Discussion** to explain how this difficulty in detecting the kinetic acceleration of dephosphorylation in the deletion mutant, compared to the blockade imposed by overexpression, reinforces our model of Rts3 as a "rheostat" rather than a binary switch.

2.1. Are the changes in phosphorylation upon deletion consistent with simple inhibition, or could Rts3 alter phosphatase specificity or Sit4 binding partner selection?

We agree that distinguishing between simple inhibition and altered specificity is crucial. While we cannot rule out subtle allosteric effects, three lines of evidence strongly argue that Rts3 acts primarily as an inhibitor of existing Sit4 complexes rather than a factor that redirects substrate specificity or alters partner selection:

- (i) If Rts3 altered specificity (directing Sit4 to new targets), we would expect *RTS3* overexpression to generate a unique phosphoproteomic signature distinct from the *sit4Δ* phenotype. Instead, we observed that 79% of Rts3-regulated sites are also regulated by *SIT4* deletion (Fig. 5a). This indicates that Rts3 does not generate "new" dephosphorylation events; rather, it suppresses a specific subset of canonical Sit4 activities.
- (ii) Our structural model and mutagenesis data show that Rts3 inserts an α -helix directly into the Sit4 catalytic cleft (Fig. 4d, e). This steric occlusion blocks substrate access to the active site, a mechanism characteristic of competitive inhibition rather than substrate re-selection.
- (iii) We show that Rts3 binds to the intact Sit4-Sap185/190 holoenzyme (Fig. 4c) rather than displacing the SAP subunits. Since SAPs are the primary determinants of Sit4 specificity, the fact that Rts3 binds the existing complex suggests it functions to "freeze" or "gate" the activity of these specific holoenzymes, rather than shuffling binding partners to create novel specificities.

Therefore, we conclude that Rts3 acts as a "molecular brake" on the Sit4-Sap185/190 branch, restricting its activity without fundamentally changing its substrate recognition properties. We have **added text in the discussion** to address this point.

2.2. How do the Rts3-dependent sites not found in the Sit4 delete fit with its mechanism of action?

We appreciate the opportunity to clarify the nature of the Rts3-unique phosphosites. While we classify these sites as "Rts3-dependent but not Sit4-dependent" based on strict statistical cutoffs, we do not believe they represent a mechanism independent of Sit4. Our conclusion is based on three key points:

- (i) **High Overlap:** The vast majority (79%) of Rts3-regulated sites are indeed Sit4-dependent (Fig. 5a), providing the primary evidence for the inhibitory model.
- (ii) **Mutant Controls:** Crucially, the Rts3^{3A} and Rts3^{13A} variants, which are specifically defective in Sit4 binding, fail to confer any rapamycin resistance or 3-AT sensitivity (Figs. 4j and new 5f). If Rts3 were acting on non-Sit4 targets to generate these "unique" phosphosites, we would expect the non-binding mutants to retain some functional impact. The fact that they are phenotypically inert strongly implies that all Rts3 outputs, including these sites, stem from the interaction with Sit4.
- (iii) **Technical and Network Factors:** The non-overlapping fraction likely arises from two sources: (a) Statistical thresholding, where a site is modulated in *sit4Δ* but falls just below the significance cutoff while passing it in *RTS3-OE*, and (b) Network feedback, where the graded "dimmer-switch" inhibition by Rts3 triggers distinct compensatory signaling loops compared to the chronic, "all-or-none" loss of *SIT4*.

Thus, these sites fit the mechanism of action as downstream consequences of modulating Sit4, rather than evidence of a separate function. To clarify this interpretation, we have **added** the following **text to the Results section**:

"The remaining Rts3-specific sites likely reflect indirect feedback or statistical thresholding rather than off-target activity, as Rts3 variants unable to bind Sit4 failed to elicit any detectable phenotypes (Figs. 4j and 5f)."

2.3. How was statistical significance defined (e.g., FDR correction, cutoffs)? Listing these thresholds would improve clarity and reproducibility.

We thank the reviewer for highlighting the need for precise statistical definitions. To ensure reproducibility and clarity, we applied rigorous thresholds for all proteomic datasets. Specifically, significance was defined using a Student's t-test with a permutation-based False Discovery Rate (FDR) of < 0.05 to correct for multiple hypothesis testing. Furthermore, for the biological stratification of targets (e.g., the Venn diagram in Fig. 5a), we additionally applied a strict fold-change cutoff of > 2 ($\log_2 > 1$) to focus on robust regulatory events. We have **corrected the legend** for Fig. 5a (which previously stated " $p < 0.05$ " rather than "FDR < 0.05") and **updated the Methods section** to explicitly list these criteria.

2.4. Additionally, the text indicates that Rts3 binds the Sit4–Sap185 complex in a way that would inhibit its activity, but this is not clearly explained. The authors should specify where the Rts3 helix binds, how this position might block or modify access to substrates, and whether the model predicts simple inhibition versus altered specificity.

We thank the reviewer for pointing out the need for greater clarity regarding the structural mechanism. Our model indicates that the Rts3 helix does not merely bind the periphery of the complex but inserts directly into the catalytic groove. Specifically, the model predicts that the Rts3 helix docks into the interface between Sit4 and Sap185, positioning itself directly over the phosphatase active site.

To rigorously validate this positioning, we calculated the Hydrogen-bond occupancy over the converged portion (100–300 ns) of our molecular dynamics simulation. This analysis revealed that Rts3 residues Asp⁷⁷ and Glu⁷³ form transient hydrogen bonds with His²³⁸, a strictly conserved residue located directly within the Sit4 active site, with the Asp⁷⁷ interaction displaying significantly higher stability (Supplementary Figs. 3a and b). This structural arrangement supports a model of simple competitive inhibition via steric occlusion. By

physically blocking the entry path for substrates and engaging directly with the catalytic center, Rts3 prevents dephosphorylation of canonical targets without creating a new binding surface for novel substrates.

We have **updated the Results** section to explicitly describe this "steric wedge" mechanism and included the H-bond occupancy data in **new Supplementary Fig. 3**.

Specific Comments and Minor Revisions

3. Line 48: Should read “A subset of these enzymes interacts with, and are regulated by...”

The subject of this sentence is "A subset" (singular). Therefore, to maintain subject-verb agreement and consistency with the first verb "interacts" (singular), we have retained the singular form "is regulated by" rather than changing it to the plural "are".

4. Line 90 and Table 1: The statement that Rts3 was “significantly enriched” in rapamycin-treated cells needs clarification. The Fig. legend just restates this fact—please specify how enrichment was determined, including criteria or statistical thresholds used (e.g., fold-change, FDR).

We agree that the statement regarding Rts3 enrichment should be supported by quantitative metrics. As detailed in Supplementary Data 1, the enrichment analysis was performed using Student’s t-tests with a permutation-based FDR < 0.05. Under these rigorous criteria, Rts3 emerged as the most significantly enriched interactor, exhibiting a log₂ fold-change of 6.53 (approximately 92-fold enrichment) in rapamycin-treated samples compared to log phase samples, and a log₂ fold-change of 3.25 (approximately 9.5-fold enrichment) in rapamycin-treated samples compared to control empty beads. We have **updated the Fig. 1 legend** to include both the statistical definition of significance and the specific fold-change value for Rts3.

5. Promoter analysis section (Fig. 2B–C): The phrase “large-scale promoter annotations and motif analyses (Saccharomyces Genome Database)” is vague. Please clarify which studies or datasets were used and cite the primary literature rather than the database itself. It is also unclear that Gat1 and Gln3 are the primary drivers of RTS3 expression. Fig. 2C suggests additional regulators may contribute, as the induction is only partially blunted in the double mutant. Replotting the data (and that of Fig. 2b) on a log₂ scale could help visualize dynamic range and clarify this point.

We appreciate the reviewer’s suggestions to improve the rigor of the promoter analysis section.

- (i) Citations: We have **replaced the generic reference** to "SGD" with a citation to the primary large-scale chromatin profiling and motif analysis that underpin these annotations (MacIsaac et al., 2006).
 - (ii) Additional Regulators: We agree that the residual expression in the *gln3Δ gat1Δ* mutant implies the existence of other regulators. As detailed in Fig. 2c, the deletion of Gln3 and Gat1 reduces *RTS3* induction by approximately 70%. While this establishes them as the primary drivers, the remaining ~30% signal suggests contributions from other pathways. As noted in our response to point 15 (reviewers #2/3; below), **we have updated the text** to explicitly acknowledge this residual expression and the potential for unidentified inputs.
 - (iii) Log₂ vs. Linear Scale: We carefully considered replotting the data on a log₂ scale. However, we opted to retain the linear (standard) scale because it intuitively conveys the "switch-like" dynamics of the induction (from near-zero to high levels). Furthermore, the linear scale clearly demonstrates that removing Gln3/Gat1 abolishes the majority (~70%) of the signal, reinforcing their role as dominant effectors while visibly retaining the residual baseline that the reviewer correctly points out.
6. TurboID dataset: With 764 significant interactors, the current analysis is limited. A more global interpretation—e.g., enrichment of functional categories or pathways—would help contextualize Rts3’s role beyond the specific Sit4-Sap185 and ribosomal interactions highlighted.

We appreciate the reviewer's suggestion to provide a broader functional context for the Rts3 proximate. While our initial analysis prioritized the Sit4-Sap185 interaction, we agree that a global interpretation is necessary to fully capture Rts3's cellular environment. To this end, we refined our analysis of the TurboID dataset, focusing on 381 high-confidence proteins significantly enriched in the Rts3 neighborhood. We performed a Gene Ontology (GO) enrichment analysis on this subset to map the global landscape of Rts3 interactions. This analysis revealed a predominant enrichment of factors involved in translation and ribosome biogenesis, providing independent support for our findings that Rts3 associates with the translation machinery during stress, in addition to its regulation of the Sit4-Sap185 module.

To visualize these findings, we performed a structural revision of Fig. 4. We integrated the specific PPase interaction network (formerly Fig. 4b) directly into **Fig. 4a** to streamline the presentation. This allowed us to **introduce a new Fig. 4b** displaying a dot plot of the global GO enrichment analysis. We also **updated Supplementary Data 2** to clearly list the 381 high-confidence interactors alongside their functional annotations. Finally, we **revised the Results section** to discuss these global functional categories, specifically highlighting the broad engagement with the translation machinery, before narrowing our focus to the specific phosphatase interactions.

7. **Structural model and transition (text around Fig. 4): The rationale for testing Rts3 as an inhibitor is not clear from the preceding description. Please explain how the predicted Rts3-binding site suggests inhibition of Sit4 activity.**

We agree that the link between the structural model and the hypothesis of inhibition should be explicit. As detailed in our response to the previous comment (regarding the structural mechanism; 2.4.), we have **expanded the description of the model** to clarify that the Rts3 helix is positioned directly over the catalytic center, acting as a steric wedge. To ensure the rationale for the subsequent experiments is perfectly clear, we have also **refined the transition** sentence leading into the *in vitro* assays to explicitly link the structural feature (occlusion) to the experimental test (inhibition).

8. **Fig. 5 and phosphoproteomic analysis: As discussed above, the logic of using Rts3 overexpression in rapamycin as the main readout is problematic, since Rts3 is already induced under these conditions. A deletion versus wild-type comparison in rapamycin would provide a cleaner view of Rts3 targets and should be emphasized in the analysis and discussion (referenced in Fig. 5E but not adequately integrated earlier).**

This comment raises the same concern regarding the phosphoproteomic experimental design (overexpression vs. deletion) as comment 2. As detailed in our response to comment 2 (above), we agree that a deletion analysis is often the gold standard for defining physiological targets. However, our analysis revealed that while *rts3Δ* results in significant downstream transcriptional and translational deregulation over time (Fig. 5e), the acute phosphoproteomic changes are subtle due to the high basal activity of Sit4 in rapamycin-treated cells. Conversely, overexpression provides a robust inhibitory signal that allows for high-confidence mapping of the direct interaction landscape. As noted in Response 2, we have addressed this by (i) **adding the *rts3Δ* phosphoproteomic dataset to Supplementary Data 3**, and (ii) **modifying the Results section** to explicitly explain the rationale for prioritizing the overexpression dataset for interaction mapping, while acknowledging the deletion data.

9. **Physiological relevance: The authors briefly mention a small delay in recovery from quiescence in *rts3Δ* cells, but this the physiological impact of the Rts3 feedback loop is not analyzed in depth. Further discussion or data addressing nutrient-limiting growth conditions (e.g., poor nitrogen sources) could significantly enhance the impact of the work.**

As this comment addresses the same physiological scope raised in Point 1, we have addressed them jointly. As detailed in Response 1, we followed the reviewer's suggestion to analyze growth under nutrient-limiting conditions (proline) and further extended the analysis to long-term survival. We agree that these additions significantly enhance the impact of the work by distinguishing between the fine-tuning of growth on poor nitrogen sources (**new Supplementary Fig. S2e**) and the critical role of Rts3 in ensuring survival during deep quiescence (**new Supplementary Fig.**

S2f). These new data and their interpretation have been **integrated into the Results and Discussion** sections as described above.

Reviewers #2 and #3:

Major Comments

1. Rts3 band patterns differ between the Fig.s 2A and 2D, especially at +AA 10 and +AA 60 minutes. Clarification or use of matched blots would improve consistency.

We thank the reviewer for this detailed observation regarding the Rts3 band patterns in Figs. 2a and 2d. We agree that visual consistency between panels is important for the interpretation of the data. To address this, we have **replaced the immunoblot in Fig. 2a** with a representative blot that better matches the exposure and degradation kinetics observed in the wild-type controls of Fig. 2d. However, we would like to offer a brief clarification regarding the inherent challenges in matching these exposures perfectly, which stems from the biological phenotype of the mutants used in Fig. 2d:

- (i) **Difference in Signal Intensity:** Fig. 2d analyzes Rts3 stability in temperature-sensitive proteasome mutants (*pup1-1* and *rpn5-1*). In these mutants, particularly *rpn5-1*, Rts3 is not degraded and accumulates to massively higher levels compared to the wild type (WT).
- (ii) **Exposure Constraints:** To resolve the accumulated, phosphorylated, and polyubiquitinated species of Rts3 in the *rpn5-1* mutant without saturating the film (which would result in a "blackout" of the signal), we are required to use a significantly shorter exposure time for Fig. 2d. Conversely, tracking the rapid physiological degradation of Rts3 in WT cells (Fig. 2a) often requires a longer exposure to visualize the residual protein at the 10- and 20-minute time points.

To illustrate this technical constraint, we have explicitly included a shorter exposure crop for the *rpn5-1* data in Fig. 2d. This highlights the dramatic difference in protein abundance and explains why the WT bands in that specific context may appear fainter or display slightly different kinetics compared to a standard degradation assay like Fig. 2a. We believe the **revised Fig. 2a**, combined with this clarification, provides a consistent and accurate representation of Rts3 dynamics

2. The slower-migrating forms of Rts3 appears only in *rpn5-1* (and with MG-132) but not in *pup1-1*. In addition, the +AA 60 band in *pup1-1* resembles wt. This inconsistency should be addressed.

We appreciate the reviewer's careful inspection of the Western blot patterns. We agree that the "slower-migrating" phosphorylated species is clearly visible in *rpn5-1* mutants and MG-132 treated cells, but largely absent in *pup1-1* cells. We attribute this to the difference in the strength of proteasome inhibition. This is internally validated by the anti-ubiquitin immunoblots (Fig. 2d), which show that *rpn5-1* and MG-132 treatment lead to a robust and rapid accumulation of polyubiquitinated proteins (with *rpn5-1* already showing elevated basal levels), whereas the *pup1-1* mutant exhibits a marked delay in ubiquitin accumulation.

Consistent with this "weaker/delayed" block, *pup1-1* is sufficient to slow bulk Rts3 turnover but retains enough residual activity to clear the specific phosphorylated intermediate. In contrast, the stringent blockade imposed by *rpn5-1* or MG-132 allows this intermediate to accumulate to detectable levels. We have **updated the text** to explicitly link the Rts3 phenotype to this difference in proteasomal efficiency.

To conclusively validate this interpretation and define the molecular mechanism, we extended our analysis to map the specific phosphodegron controlling this process. We identified that a single point mutation at Ser¹¹¹ (S111A) is sufficient to fully stabilize Rts3 during amino acid refeeding. Crucially, this stabilized Rts3^{S111A} mutant migrates exclusively as the faster species and lacks the slower-migrating isoform observed in the proteasome mutants. This genetic evidence confirms that the "upper band" noted by the reviewer is indeed the specific, phosphorylated transition state required for SCF^{Cdc4}-mediated degradation. We have integrated these findings into the Results section and **updated Fig. 2i** to showcase the stabilization of the Rts3^{S111A} mutant.

3. Including *TOR1-1* cells in Fig. 2E would demonstrate that the mobility shift is TORC1-dependent rather than an artifact of rapamycin exposure.

We thank the reviewer for this excellent suggestion. To rule out any potential off-target effects of rapamycin, we monitored Rts3 degradation in cells expressing the rapamycin-resistant *TOR1-1* allele. We performed the refeeding experiment in *TOR1-1* cells in the presence of rapamycin. As expected, while rapamycin stabilized Rts3 in wild-type cells by inhibiting TORC1, it failed to do so in *TOR1-1* cells. In this background, Rts3 was rapidly degraded, behaving exactly like the untreated control. This confirms that the stabilization observed in wild-type cells is indeed due to the specific inhibition of TORC1 activity and not an artifact of the drug. We have included these data as **new Fig. 2g** in the revised manuscript.

4. Ubiquitinated-protein blots differ between Fig.s 2D and 2F and should be more comparable.

We assume the reviewer is referring to the comparison between Fig. 2d and Fig. 2e, as Fig. 2f does not contain an anti-ubiquitin immunoblot. We appreciate the reviewer pointing out this inconsistency. We have addressed this concern through both a technical correction to the Fig. and a clarification of the biological context:

- (i) Technical Correction (Fig. 2e replacement): We recognized that the anti-ubiquitin blot originally presented in Fig. 2e was overexposed. This resulted in visible high-molecular-weight smears even in the vehicle-treated control samples, creating an apparent discrepancy with the cleaner control lanes in Fig. 2d. To resolve this, we **have replaced the anti-ubiquitin blot in Fig. 2e** with a shorter exposure of the same experiment. In this revised panel, the control samples now show negligible accumulation of ubiquitinated proteins, making the baselines in Fig. 2d and 2e fully comparable.
- (ii) Biological Context: With the controls now consistent, the remaining differences in the experimental lanes reflect the specific "strength" of the proteasomal blockade. Fig. 2d compares *pup1-1* (a leaky/hypomorphic 20S allele) with *rpn5-1* (a tight 19S allele). Consequently, ubiquitin accumulation is delayed/moderate in *pup1-1* but massive in *rpn5-1*. The robust accumulation seen in Fig. 2e (MG-132 treatment) mirrors the strong blockade of *rpn5-1* rather than the partial phenotype of *pup1-1*.

We believe the **replacement of the blot in Fig. 2e**, combined with this explanation, ensures the data are visually consistent and biologically accurate.

5. More methodological detail on the kinase assay should be added in the main text and Fig. legend, and the authors may wish to comment on the reason why they used wortmannin instead of rapamycin as in the rest of the manuscript for the same purpose.

We appreciate the reviewer's request for clarification regarding the inhibitor choice. We have updated the manuscript to include the specific experimental parameters of the kinase assay. Regarding the use of wortmannin: Rapamycin is an allosteric inhibitor that strictly requires the immunophilin cofactor FKBP12 (Fpr1 in yeast) to bind the TORC1 FRB domain. Since Fpr1 is not a stable subunit of the purified TORC1 complex, rapamycin is ineffective in this purified system. Therefore, we used wortmannin, which has been established as a direct inhibitor of the TORC1 catalytic domain in the absence of FKBP12 (Brunn et al., 1996).

We **expanded the description in the Results** section to specify the purification sources and reaction conditions. We **updated the Methods** to explicitly state that wortmannin was used to inhibit TORC1 activity as previously described (citing Brunn et al., 1996), noting that the absence of FKBP12 prevents the use of rapamycin in this assay.

6. Including described Rts3 interactors as control in Fig. 3A would confirm that the absence of interaction with Tpd3 and Pph21/22 is not due to low Rts3 abundance.

We agree that confirming the abundance and functionality of the Rts3 bait is essential. We believe this control is provided by the combination of Fig. 3a and Fig. 4c:

- a. Abundance: The anti-GFP blot in Fig. 3a (IP lanes) explicitly demonstrates that GFP-Rts3 was immunoprecipitated at high levels, comparable to the positive controls (Rts1 and Cdc55). This directly rules out the possibility that the absence of Tpd3/Pph21/22 co-purification is due to insufficient bait protein on the beads.
- b. Functionality: We show in Fig. 4c that the exact same GFP-Rts3 construct, under rapamycin-treated conditions, robustly co-immunoprecipitates Sit4, Sap185, and Sap190.

To address the reviewer's concern without disrupting the sequential Fig. numbering in the manuscript, we have **updated the text** to explicitly highlight the efficient immunoprecipitation shown in Fig. 3a **and textually reference** the functional binding described in the subsequent section.

7. The experimental procedure for Fig. 4c requires further explanation. The Rts3-3A mutant shows differential loss of interaction with Sit4, Sap185 (clear decrease) compared with Sap190 (not so clear decrease); this difference should be addressed and clarified.

We agree that the differential impact of the Rts3^{3A} mutation on Sap185 versus Sap190 binding warrants further explanation. We verify that the Rts3^{3A} mutations (R65A, E73A, D77A) target residues specifically required for anchoring Rts3 to the Sit4 catalytic cleft. Since Sit4 is the common catalytic subunit in both Sap185 and Sap190 complexes, one might expect an identical loss of binding if the interaction were solely mediated by Sit4. However, we observe that while Rts3^{3A} loses binding to the Sit4-Sap185 complex, it retains significant binding to the Sit4-Sap190 complex. This differential effect suggests that the Sit4-Sap190 complex possesses additional structural determinants, likely direct contacts between Rts3 and Sap190, that are absent in the Sit4-Sap185 complex.

These "secondary" contacts in Sap190 appear sufficient to tether Rts3 to the complex even when the primary "catalytic cleft" anchor (mediated by the 3A residues) is disrupted. Importantly, despite this physical retention, the Rts3^{3A} mutant is functionally dead (Fig. 4j and new Fig.5f), confirming that the specific engagement with the Sit4 active site is the critical inhibitory mechanism.

To clarify this distinction, we have **added the following text** to the Results section:

"Since the Rts3^{3A} mutation disrupts the Sit4 interface (common to both complexes), the residual binding to Sap190 suggests that Rts3 maintains secondary, Sap190-specific contacts that stabilize the complex to some extent even when the primary active-site anchor is compromised."

8. The manuscript uses 4, 5 and 10 ng/ml rapamycin in different assays. A brief justification would help. Growth differences between Fig. 4J (5 ng/ml) and Fig. 5G (10 ng/ml) should be clarified. In Fig. 4J first spot in rapamycin plate is growing weakly at 5ng/ml but in Fig. 5G at 10ng/ml authors present better growth.

We thank the reviewer for identifying the different rapamycin concentrations. As noted, yeast cells are significantly more sensitive to rapamycin in synthetic defined (SD) media than in rich (YPD) media. Therefore, we empirically titrated the concentrations for each assay to optimize the dynamic range:

- a. Fig. 4j (SD-Ura): We used 5 ng/mL because cells are hypersensitive in synthetic selection media.
- b. Fig. 5g (YPD): We used 10 ng/mL because cells in rich media are more robust, requiring a higher dose to clearly distinguish the rescue phenotypes of suppressor mutants.

We have added a **clarifying sentence to the Methods** section specifying that rapamycin concentrations were adjusted according to the media composition.

9. The phenotype observed on 3-AT plates in Fig. 5F is quite subtle. Including a control strain to verify that the 3-AT assay is working effectively would support the conclusion. The authors may also wish to adjust the wording accordingly.

We thank the reviewer for this constructive suggestion. We agree that the phenotype of the wild-type *RTS3* overexpression strain on 3-AT is subtle. To rigorously verify the effectiveness of the assay and the specificity of the observed phenotype, we have updated Fig. 5f to include a *gcn4Δ* strain as a positive control for 3-AT sensitivity. As expected, the *gcn4Δ* mutant is unable to grow on this medium, confirming that the assay conditions effectively monitor Gcn4-dependent readouts. Furthermore, we assessed cells overexpressing the Sit4-binding defective variants *rts3^{33A}* and *rts3^{13A}*. Unlike the wild-type *RTS3* overexpression, neither of these variants caused sensitivity to 3-AT. This crucial control demonstrates that the dampening of the GAAC pathway is not an artifact of overexpression, but strictly depends on the ability of Rts3 to engage and inhibit the Sit4 phosphatase. We have **adjusted the text and Fig. 5f** accordingly.

10. The genetic logic may not be intuitive to general readers. A more detailed explanation will benefit the manuscript. Authors could consider to use an additional experimental approach to show the same, which would strengthen the conclusion.

We appreciate the reviewer's suggestion to improve the clarity of the genetic analysis. We interpret this comment as referring to the epistasis experiments in Fig. 5g, where the "rescue" of *rts3Δ* sensitivity by deletion of *SIT4* or *GLN3* relies on the logic of relieving pathway hyperactivation. To address this, we did the following.

- a. Clarified logic: We have **expanded the text in the Results** section to explicitly articulate the logical premise: if the sensitivity of *rts3Δ* is caused by unrestrained (toxic) Sit4 activity, then removing Sit4 should cure the sensitivity.
- b. Complementary Approach: Regarding the request for an additional approach, we highlight that we have effectively performed the reciprocal "gain-of-function" experiment in Fig. 3b. There, we show that *RTS3* overexpression confers rapamycin resistance, the exact inverse of the deletion phenotype.

Together, these two genetic approaches, loss-of-function causing sensitivity (rescued by downstream deletions) and gain-of-function causing resistance, provide robust, orthogonal evidence that Rts3 functions as a negative regulator.

11. In the Discussion, the statement that Rts3 "counteracts TORC1-dependent phosphorylation of the hibernation factor Stm1 that balances ribosome homeostasis" implies enhanced Stm1 dephosphorylation, which seems at odds with the model in which Rts3 inhibits Sit4, the phosphatase expected to act on Stm1. This apparent contradiction should be addressed, unless this reviewer misunderstood.

We thank the reviewer for catching this apparent contradiction. Indeed, the statement that Rts3 "counteracts TORC1-dependent phosphorylation" was semantically imprecise and inadvertently implied that Rts3 promotes dephosphorylation. This contradicts our established model (and phosphoproteomic data) where Rts3 acts as an inhibitor of Sit4. Mechanistically, Sit4 is responsible for removing the inhibitory phosphates from Stm1. By inhibiting Sit4, Rts3 prevents this dephosphorylation, thereby preserving the phosphorylated species (or slowing its turnover). We have **corrected the Discussion** text to accurately reflect this mechanism.

Minor Comments

12. Strain names should be included in the Fig. legends to improve clarity of the experimental details.

We appreciate the reviewer's concern for clarity and reproducibility. We agree that knowing the exact genetic background is essential for every experiment. However, inserting specific laboratory strain identifiers (e.g., "MP6172", "YL515") directly into the Fig. legends would significantly disrupt the flow and readability of the text. To ensure rigorous traceability without

compromising clarity, we have provided a comprehensive list of all strains, including their exact genotypes and the specific Fig. panels in which they are used, in **Supplementary Table 1**.

This approach allows readers to immediately identify the biological context (e.g., *rts3Δ*, WT) in the Fig. legends while easily retrieving the precise technical details from the comprehensive table, consistent with Nature Communications standards for data presentation.

13. As Fig. 1B supports the proposed negative feedback loop, explicitly referring back to it when this mechanism is discussed later in the text would help the reader.

We agree that explicitly linking the initial discovery (Fig. 1b) to the final mechanistic model helps close the narrative loop for the reader. Fig. 1b provides the unbiased proteomic evidence that Rts3 is robustly enriched in the phosphatase interactome upon TORC1 inhibition. This enrichment is the direct consequence of the Gln3/Gat1-mediated induction described in our feedback model. We have **added a cross-reference** to Fig. 1b in the **Discussion section**, specifically where the transcriptional induction component of the feedback loop is described.

14. Page 4, line 113 onwards: a more detailed explanation would help readers follow Fig. 1C. In addition, it may be worth reconsidering whether the term 'genetic analysis' is the most appropriate description in this context.

We appreciate the reviewer's suggestion to improve the clarity of the initial results section.

- a. Explanation of Fig. 1c: We agree that the description of the protein interaction network was too brief. We have **added a sentence** to the Results section to explicitly explain that Fig. 1c represents a STRING-based network visualization, which serves to highlight the recovery of intact, distinct phosphatase holoenzymes (rather than just isolated subunits) in our screen.
- b. Terminology ("Genetic Analysis"): We agree that the term "genetic analysis" (used to describe the identification of Gln3/Gat1) is imprecise in this context, as it implies a forward genetic screen rather than the testing of specific candidates derived from promoter analysis. We have **replaced this term with "targeted deletion analysis"** to accurately reflect the experimental approach.

15. Fig. 2C: as residual *RTS3* expression persists in *gat1 gln3* deleted cells, the authors may wish to discuss additional pathways that could contribute to *RTS3* expression.

We agree that the residual *RTS3* expression observed in the *gln3Δ gat1Δ* double mutant (Fig. 2c) indicates that other factors may contribute to its regulation, likely maintaining basal levels or providing minor inputs. To address this, we systematically tested the most prominent candidates predicted by promoter motif analysis, including Gcn4 (General Amino Acid Control), Rtg3 (Retrograde signaling), and Msn2/4/Gis1 (General Stress Response), but found that their deletion did not significantly alter *RTS3* profiles (Fig. 2c). Consequently, while we cannot identify the specific source of this residual expression, we conclude that it is independent of these major stress pathways. We have **updated the Results** section to explicitly acknowledge this residual expression and the possibility of secondary, unidentified regulatory inputs.

16. Fig. 3B: showing *Rts3* levels expressed from the endogenous promoter and from *ADH1* promoter could help define the expression threshold above which cells become rapamycin-insensitive.

We appreciate this insightful suggestion. Defining the protein abundance relative to the observed phenotype adds important context to the genetic data. As requested, we have **updated Fig. 3b to include immunoblots** for GFP-Rts3 (and Adh1/2 loading controls) directly alongside the spot growth assays. These blots confirm that expression from the *ADH1* promoter results in substantially higher protein levels (in exponentially growing cells) compared to the endogenous promoter. This direct comparison clarifies that the rapamycin-resistance phenotype is strictly dosage-dependent, requiring Rts3 accumulation well above physiological levels to effectively antagonize TORC1 inhibition.

17. Fig. 3D: larger crops of cells would improve visibility of the phenotypes shown.

We agree that larger crops significantly enhance the visibility of the subcellular localization patterns. Accordingly, we have **updated Fig. 3d to include magnified insets** for each condition. These insets show the regions marked by white dotted squares in the merged panels, providing a clearer and more detailed view of the cytoplasmic-to-nuclear transition of Rts3.

18. Fig. 4E: some residues are difficult to see; alternative colouring may enhance clarity.

To enhance clarity, we **have revised Figs. 4e and 4f** by adding a white background behind the amino acid labels, ensuring that the residue numbers remain clearly legible against the structural features.

19. Page 34, line 897 (Fig. 4): please clarify why “not shown”.

We apologize for the lack of clarity. We originally used the phrase “not shown” because Fig. 4e presents a static structural model, which made it impossible to visualize the dynamic and alternating nature of the hydrogen bonds involving the catalytic residue His²³⁸. To address this limitation and provide quantitative evidence, we performed a hydrogen-bond occupancy analysis over the converged trajectory of our MD simulation. This analysis reveals that His²³⁸ forms transient hydrogen bonds primarily with Asp⁷⁷, and less frequently with Glu⁷³, confirming that these residues dynamically engage the active site.

We have included this quantitative data as **Supplementary Figs. 3a and b** and have updated the manuscript text to reference this figure instead of stating “not shown.”

20. Page 34, line 912 (Fig. 4): clarify whether expression was tested in asynchronously growing WT cells.

We confirm that the immunoblot analysis presented in Fig. 4j (lower panel) was performed using lysates from asynchronously growing cells (exponential phase). This control was included to verify that WT and mutant Rts3 variants are overexpressed to comparable levels under standard growth conditions, before stress assays. We have **updated the legend of Fig. 4j** to explicitly state that protein expression was assessed in “asynchronously growing cells,” ensuring the experimental context is unambiguous.

21. Page 8, line 211: additional detail on the phosphatase assay should be added to the main text.

We agree that providing specific experimental parameters in the main text improves clarity and reproducibility. We have expanded the description of the *in vitro* phosphatase assay in the Results section to explicitly state the source of the protein components (immunopurified Sit4 vs. recombinant Rts3) and the identity of the substrate used.

We have **revised the relevant paragraph** in the Results section to include these technical details. The text now reads: “...we incubated HA-Sit4 (immunopurified from yeast) with increasing concentrations of recombinant His₆-Rts3 (purified from *E. coli*) and assessed phosphatase activity against the synthetic phosphopeptide substrate KRpTIRR.”

22. Editorial Points

Page 49: rephrase “Tip1-mediated Tap42 sequestering”.

Line 229: “2720” rather than “2’720”.

Line 236: clarify the meaning of “network-level dissection”.

Line 242: specify the protein in which Ser41 and Ser45 are affected.

Line 254: clarify “shared Sit4- and Rts3-regulated nodes” for readers unfamiliar with STRING.

Line 271: clarify “permease regulatory”.

Line 280: rephrase the “dual-mode regulation–feedforward induction” sentence.

Line 807: correct “declaration od”.

We have amended the manuscript to incorporate all listed editorial corrections, clarifications, and rephrasings

Reviewer #4:

Major Comments

1. The authors claim that Rts3 is a direct substrate of TORC1 by performing an *in vitro* kinase assay (Fig. 2G). Have the authors confirmed the physical interaction between TORC1 and Rts3, which would further support their claim?

We thank the reviewer for this insightful comment. We agree that demonstrating a physical interaction between TORC1 and Rts3 significantly strengthens the conclusion that Rts3 is a direct substrate. Capturing this interaction is inherently challenging for two reasons. First, as shown in Fig. 2, the interaction triggers immediate phosphorylation and degradation of Rts3. Second, kinase-substrate interactions are typically transient ("kiss-and-run") and sensitive to the lysis and washing steps required for Co-IP. Even when protein levels are stabilized, the physical complex often dissociates during purification.

To address the reviewer's request, we therefore chose the Yeast Two-Hybrid (Y2H) system. This assay detects interactions *in vivo*, avoiding the disruption caused by cell lysis, and its genetic readout (growth) allows for the accumulation of signal from transient binding events. By combining this sensitive assay with the proteasome inhibitor MG-132 to prevent substrate turnover, we were able to capture the specific physical interaction between the TORC1 subunit Tco89 and Rts3 (**Supplementary Fig. 1**). This result, combined with the *in vitro* kinase assay (Fig. 2h), provides compelling evidence that Rts3 physically associates with the TORC1 complex and is a direct substrate.

2. In Fig. 3B, the authors shows that the overexpression of Rts3 from ATD promoter confers rapamycin resistance on budding yeast cells. Since this experiment relies on the Rts3 overexpression, the observed rapamycin resistance might not reflect Rts3-mediated inhibition of Sit4, but could *instead* be an artifact of overexpression. To rule out this possibility, the same experiment should be performed using *sit4* deletion mutant cells, in which rapamycin resistance would not be expected.

We followed the reviewer's suggestion and performed the experiment in a *sit4Δ* background. As predicted, the rapamycin resistance conferred by *RTS3* overexpression was completely abolished in the absence of *SIT4*, confirming that Rts3 acts through the phosphatase (**new Figure 4j**). Importantly, we observed that the *sit4Δ* mutant itself was not rapamycin resistant and behaved like WT. This result aligns with previous work (Jablonski et al., 2009) showing that rapamycin resistance is a phenotype specific to the loss of the Sap185/190 regulatory subunits, whereas the combined loss of Sap4/155 or the catalytic subunit Sit4 does not confer resistance. Therefore, the fact that *RTS3* overexpression confers resistance strongly argues that it is not a generic overexpression artifact or a global inhibitor of Sit4 (which would phenocopy *sit4Δ*). Instead, Rts3 functions as a specific functional phenocopy of the *sap185Δ sap190Δ* deletion, consistent with our physical data showing Rts3 binds the Sit4-Sap185/190 complex. We have **updated the Results** section to include these findings in **new Figure 4j**.

3. The authors validated the interaction model of Sit4 and Rts3 obtained from MD simulations by co-immunoprecipitation assays with alanine mutants of Rts3 (Fig. 4C, E and F). On the other hand, the importance of Sit4 residues R266, Y263, and Y124, which have been proposed to interact with Rts3, has not been experimentally validated. It would be valuable to assess the significance of these residues using co-immunoprecipitation (co-IP) with alanine-substituted Sit4 mutants.

We thank the reviewer for this suggestion and attempted to validate the interface from the Sit4 side by generating the requested alanine substitutions (R266A, Y263A, and Y124A). However, strains expressing these Sit4 variants from CEN/ARS plasmids were extremely sick and could not be analyzed reproducibly in our signaling assays, as noted above. Based on our structural model,

these residues lie within the catalytic center of Sit4 (Fig. 4e), and their mutation is therefore likely to alter overall phosphatase activity rather than selectively affect the interaction with Rts3. As a result, the observed phenotypes are difficult to interpret in terms of interface specificity. In contrast, the Rts3 mutants, shown in Fig. 4c, 4h, 4j, and 5f, selectively perturb the interaction while preserving Sit4 catalytic function, and we therefore consider them to provide the most informative validation of the interface.

4. In Fig. S1C, the growth delay observed in *rts3Δ* and GFP-RTS3-OE cells may not necessarily reflect a slower recovery from nitrogen starvation compared to wild-type cells, but could instead be due to an overall slower growth rate. This possibility could be addressed by examining growth curves using cells in logarithmic phase.

We thank the reviewer for this insightful observation. Prompted by this suggestion, we performed a comprehensive re-evaluation of both logarithmic growth rates and post-starvation recovery kinetics with increased biological replicates. Upon this rigorous side-by-side comparison, we found that the differences in regrowth kinetics between the wild-type and *rts3Δ* or *RTS3*-OE strains were not statistically robust when fully normalized against logarithmic growth rates. To ensure the highest degree of accuracy and reproducibility, we have removed the claims regarding "delayed regrowth" from the revised manuscript. However, this re-evaluation was extremely valuable as it redirected our focus toward the stability of the quiescent state rather than the kinetics of exit. As detailed in our response to **Point 1 of reviewer 1** and incorporated into the revised text:

- We found that Rts3 is critical for long-term survival during starvation. New chronological lifespan (CLS) assays show that *rts3Δ* cells exhibit significantly reduced viability over time.
- We identified a role for Rts3 in fine-tuning growth on poor nitrogen sources. We now show that Rts3 overexpression causes a growth defect on proline (likely via Sit4 inhibition), whereas *rts3Δ* cells show a modest advantage.

We believe these new data (now presented in **Supplementary Fig. 2e and 2f**) provide a more robust and physiologically relevant definition of Rts3 function - specifically, that it acts as a "brake" to prevent excessive metabolic commitment during scarcity and ensure survival, rather than regulating the speed of recovery.

5. In the polysome analysis shown in Fig. S1D, no statistically significant difference was observed between wild-type and *rts3Δ* cells. Moreover, although the difference between wild-type and GFP-Rts3-OE cells is statistically significant, it is rather small, as noted by the authors. Nevertheless, as shown in Fig. S1C, both *rts3Δ* and GFP-RTS3-OE cells exhibit a similar delay in recovery from nitrogen starvation compared to wild-type cells. Therefore, I think that the authors' conclusion 'Together, these results confirm that Rts3 associates and constrains Sit4 on hibernating ribosomes, allowing efficient recovery from prolonged starvation, thus emphasizing its physiological relevance (lines 249-251)' is not supported by the data.

We thank the reviewer for this careful observation. We agree that the differences in the polysome profiles (Fig. S2c) are subtle and do not, on their own, fully explain the recovery defects previously observed. However, we believe the physiological relevance of the Rts3-Sit4 interaction on ribosomes is supported by the phosphoproteomic state of the hibernation factor Stm1, rather than the global stability of the 80S complex.

- (i) Stm1 Phosphorylation: As shown in our phosphoproteomics dataset (Supplementary Data 3) and discussed in the text, Stm1 phosphorylation (specifically Ser⁴¹ and Ser⁴⁵) is sensitive to Rts3 levels.
- (ii) The "Rheostat" Mechanism: In *rts3Δ* cells, Sit4 is unrestrained, leading to a trend of Stm1 dephosphorylation. In *RTS3*-OE cells, Sit4 is blocked, preventing Stm1 dephosphorylation.
- (iii) Phenotypic Outcome: We propose that both extremes, hyperactive dephosphorylation (*rts3Δ*) and blocked dephosphorylation (*RTS3*-OE), disrupt the precise timing required for efficient exit from quiescence.

The fact that the 80S peak remains stable in *rts3Δ* cells (unlike in *stm1Δ* cells where it collapses) indicates that Rts3 is not a structural component required for ribosome assembly, but a regulator that "fine-tunes" the signaling state of the hibernation complex. To address the reviewer's concern and avoid overstating the polysome data, we have revised the concluding sentence to be more precise:

Revised Text:

"Together, these results confirm that Rts3 associates with hibernating ribosomes and regulates the phosphorylation status of the hibernation factor Stm1. This suggests that Rts3 fine-tunes Sit4 activity on ribosomes to ensure efficient recovery from prolonged starvation."

Minor points

6. In the input samples of Fig. 4C, a strong non-specific band is observed in addition to GFP-Rts3, whereas no such band appears in Fig. 4A. What accounts for this discrepancy?

We assume the reviewer is referring to Fig. 3a, as Fig. 4a presents mass spectrometry data. We note that an additional band is visible in the Input of Fig. 4c that is not observed in Fig. 3a. While the precise origin of this band cannot be definitively assigned, several experimental differences between these figures should be considered. Although both experiments were performed using native cell lysis, the buffer compositions differed between Fig. 3a and Fig. 4c, which can influence background signals in immunoblot analyses. In addition, the treatment conditions were not identical: Fig. 3a shows samples treated with rapamycin alone, whereas Fig. 4c includes a combination of rapamycin treatment and nitrogen starvation. Finally, the immunodetection strategies differed between the two experiments. In Fig. 4c, light chain-specific secondary antibodies were used for immunoblot detection (see Supplementary Table 1), minimizing interference from IgG heavy chains (~50 kDa) and improving specificity of the co-immunoprecipitation signals. **This detail has now been explicitly stated in the legend of Fig. 4c.** In contrast, Fig. 3a employed GFP-Trap-based detection, for which light chain-specific secondary antibodies were not required. Together, these experimental differences provide a plausible explanation for the appearance of the additional band in Fig. 4c.

7. part of the immunoblotting image in Fig. 4C (right side) appears to be missing.

We appreciate the reviewer's careful observation, and the complete immunoblot for Fig. 4C is now shown.

8. In Fig. 4E and F, the red numbering of the amino acid residues overlaps with the structure colored in magenta, making it difficult to distinguish.

To enhance clarity, we have **revised Figs. 4e and 4f** by adding a white background behind the amino acid labels. This ensures that the residue numbers are clearly legible against the structural details

9. Statistical analysis should be performed for Fig. 4H.

We have **updated Fig. 4h** to indicate statistical significance (using asterisks) for comparisons between HA-Sit4 without inhibitor and HA-Sit4 in the presence of wild-type or mutant Rts3. Accordingly, the Fig. legend has been revised to specify the statistical test and significance thresholds (n=3, ±SD; unpaired two-tailed t-test vs. HA-Sit4 without inhibitor; *p < 0.05, **p ≤ 0.01, ***p ≤ 0.001, ****p ≤ 0.0001).

10. In line 125, 'In' should not be capitalized

This typographical issue has now been addressed.

Response to Reviewer Comment

Reviewer Comment: *“Suggest that the authors spend a bit more time editing the discussion as some of the english still reads like a draft. For example, three sentences in lines 364 to 372 start with the word crucially, and the flow in this section could be improved (sentence to sentence linkages etc).”*

Response: We thank the reviewer for this helpful observation. We have carefully revised the Discussion section to improve the narrative flow and sentence-to-sentence linkages. Specifically, we have removed the repetitive use of the word "Crucially" in the indicated paragraph and replaced it with more varied logical connectors (e.g., "Importantly," "Furthermore," "In this context").

Additionally, we conducted a thorough stylistic polish of the surrounding text to remove draft-like phrasing (such as "The fact that...") and ensure the argument flows cohesively from the molecular mechanisms to the biological phenotypes. We believe these changes have significantly improved the readability and professional tone of the section.